# A proposal to extract and enhance four-Majorana interactions in hybrid nanowires

Tasnum Reza, Sergey M. Frolov and David Pekker

Department of Physics and Astronomy, University of Pittsburgh, 15260, USA

## Abstract

We simulate the smallest building block of the Sachdev-Ye-Kitaev (SYK) model, a system of four interacting Majorana modes. We propose a 1D Kitaev chain that has been split into three segments, i.e., two topological segments separated by a non-topological segment in the middle, hosting four Majorana Zero Modes at the ends of the topological segments. We add a non-local interaction term to this Hamiltonian which produces both bilinear (two-body) interactions and a quartic (four-body) interaction between the Majorana modes. We further tune the parameters in the Hamiltonian to reach the regime with a finite quartic interaction strength and close to zero bilinear interaction strength, as required by the SYK model. To achieve this, we map the Hamiltonian from Majorana basis to a complex fermion basis, and extract the interaction strengths using a method of characterization of low-lying energy levels and then finding the differences in energies between odd and even parity levels. We show that the interaction strengths can be tuned using two methods - (i) an approximate method of tuning overlapping Majorana wave functions (without non-local interactions) to a zero energy point followed by addition of a non-local interaction, and (ii) a direct parameter space optimization method using a genetic algorithm. We propose that this model could be further extended to more Majorana modes, and show a 6-Majorana model as an example. Since eigenspectral characterization of one-dimensional nanowire devices can be done via tunneling spectroscopy in quantum transport measurements, this study could be performed in experiment.

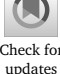

# 1 Introduction

## 1.1 Our goal

Our goal is to design the building block of the Sachdev-Ye-Kitaev (SYK) model realizabe in an experimental setup, i.e., a single four-Majorana Zero Mode interacting system. This is inspired by the idea of an experimental device that implements the SYK Hamiltonian [1]. However, our motivation is not to build an SYK model, rather it is to detect the presence of four-body Majorana interactions in experiments, and to construct a method which enables us to enhance and extract the interaction strength in an experimental setup. The key initial steps to constructing such a device are: (1) finding the experimental signatures of a quartic interactions between the MZMs and (2) tuning the device to minimize bilinear terms coupling pairs of MZMs with respect to four-body interaction strength. Here, we theoretically consider a simple setup composed of a Kitaev chain nanowire hosting four MZMs, all interacting with each other via two-body and four-body interactions. First and foremost we develop a method of characterizing the interaction strengths using the eigenspectrum of our Hamiltonian. Next, we explore the conditions under which we can enhance the quartic interactions and suppress the bilinear interactions. Finally, looking ahead but not as our main focus, we consider how to scale up the setup to N=6 Majoranas.

## 1.2 Context

### 1.2.1 What is the SYK model?

SYK model is a 0+1 dimensional model of $N_\gamma$ MZMs with random, all-to-all four body interactions. Kitaev [1] proposed this Majorana-based model as a variant of the original SY model of Sachdev and Ye [2] that described spins with random all-to-all couplings. The Hamiltonian for this model is:

$$H = \sum_{i<j<k<l}^{N_\gamma} J_{ijkl} \gamma_i \gamma_j \gamma_k \gamma_l, \tag{1}$$

where $\gamma_i$'s are the Majorana operators that obey the canonical anti-commutation relations:

$$\{\gamma_i, \gamma_j\} = \delta_{ij}, \qquad \gamma_i^\dagger = \gamma_i. \tag{2}$$

$J_{ijkl}$ are independent and identically distributed (i.i.d.) real random numbers drawn from a Gaussian distribution with zero mean and standard deviation given as:

$$\overline{J_{ijkl}^2} = \frac{3!J^2}{N_\gamma^3}, \qquad \overline{J_{ijkl}} = 0. \tag{3}$$

### 1.2.2 Significance of the SYK model

There are many aspects that make the SYK model particularly interesting [1, 3, 4]. It is a strongly interacting model with symmetry properties that closely resemble quantum gravity. A remarkable property of this model is that it is maximally chaotic in the large $N_\gamma$ limit. Black holes scramble information at a fast rate, i.e., they are maximally chaotic, characterized by the upper limit of the Lyapunov exponent. It has been shown theoretically that the four-point out of time ordered correlators (OTOC) [5] in the SYK model also saturate the upper bound on the Lyapunov exponent [6]. Thus, from a quantum chaos perspective the SYK model mimics the physics of black holes, i.e., it has holographic duality to black hole physics [7]. Besides this, the SYK model being exactly solvable at large $N_\gamma$ limit presents itself as a great tool towards simplifying understanding of the physics of quantum gravity in general. All these combined makes SYK the perfect toy model for engineering experimentally realizable black hole models.

### 1.2.3 Theoretical proposals towards experimental SYK devices

There have been a couple of proposals in the recent past that address challenges towards a practically realizable Sachdev-Ye-Kitaev (SYK) device which include (i) designing a system with a large number of MZMs with random interactions and (ii) formulating a method to suppress unwanted bilinear interactions that come along with the quartic interaction in a large-$N_\gamma$ system. One set of proposals features multiple nanowires coupled via a central quantum dot [8,9], where they use a time reversal symmetry argument to suppress the bilinear terms. In another proposal, MZMs are coupled around a vortex by engineering a nanoscale hole in the superconducting film on the surface of a three-dimensional topological insulator [10]. At the neutrality point, it is hypothesized that the bilinear terms are suppressed. A graphene based model [11] has also been put forward. Other than condensed matter systems, there are have been attempts to realize SYK in ultracold atoms [12], optical lattice systems [13], nuclear spins [14]. Digital quantum computers were also proposed to simulate the SYK model [15–17].

## 2 The two-complex-fermion model

In this section we show that by characterizing the energy level spacings between the odd and even parity states we can back out the interactions that are present in the system. This is the first step for tuning the system to the SYK point which also requires control over the interaction strength (see Section 4).

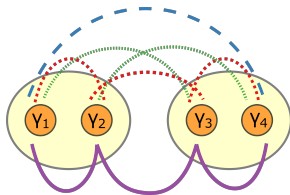

Figure 1: Two fermion sites (ovals) with four Majoranas $\gamma_1$-$\gamma_4$ (circles). All possible bilinear terms are shown as lines above, and the single quartic term as the line below the circles.

Our starting point is the two-complex-fermion model that hosts four MZMs. We analyze this model with the goal of establishing the relationship between the fermionic spectrum that can be probed in transport experiments and the interaction terms in MZM Hamiltonians.

The model consists of two fermion operators, each of which can be decomposed into a pair of MZMs. In terms of the four MZM operators $\gamma_1-\gamma_4$, the interactions between the four MZMs can be of two types - (i) six different two-body, or bilinear, interactions $K_{ij}$ and (ii) a single four-body , or quartic, interaction $J_{1234}$ that includes all four MZMs in this model (Fig. 1). The Hamiltonian of this model is:

$$H = J_{1234}\prod_{i=1}^{4}\gamma_i + i\sum_{1\leq i<j\leq 4}K_{ij}\gamma_i\gamma_j\,. \tag{4}$$

The representation that we have used for $\gamma$'s in this paper is described in Appendix A.

At the same time, the spectrum of the two-complex fermion system can also be described by the Hamiltonian:

$$H = \lambda_1(2n_1-1) + \lambda_2(2n_2-1) - u(2n_1-1)(2n_2-1), \tag{5}$$

where $n_1 = (f_1^\dagger f_1)$ and $n_2 = (f_2^\dagger f_2)$ are the quasiparticle number operators, $\lambda_1$ and $\lambda_2$ are the quasiparticle energies and $u$ is the interaction strength of the two quasiparticles (and we have

dropped the constant term as we are only interested in energy differences). We note that there are multiple different complex fermion Hamiltonians that result in the same spectrum that are connected by means of Bogoliubov transformations. We choose the representation of Eq. (5) because this Hamiltonian is in the diagonal form. We establish the relation between the MZM and the complex fermion representations of the Hamiltonian in Appendix B.

## 2.1 Energy spectra

In this section, we investigate the effect of the bilinear and quartic interaction terms on the spectral properties of the Hamiltonian. From the complex fermion representation Eq. (5), it is clear that the Hamiltonian is parity preserving with four energy levels, a pair of odd parity and a pair of even parity states. Switching focus to the MZM representation Eq. (4), we observe that tuning the bilinear and the quartic interaction terms results in a shift of the even and odd energy levels. Crucially, the shifts in the energy levels depend on which terms were tuned. In Fig. 2 we show some examples of how tuning terms in the MZM representation of the Hamiltonian affects the energy level spacings between odd (shown with black dashed lines) and even (shown with red solid lines) parity states. We note that in Fig. 2 we label the states by their complex fermion occupation numbers using the connection between representations from Appendix B.

In the top row we show the plots for the dependence of the energy levels on $J_{1234}$ with $K_{ij}$ fixed. In Fig. 2(a), we set $K_{ij} = 0$, and observe the splitting of states of different parity as a function of $J_{1234}$, while the same parity states stay degenerate throughout. This shows that $J_{1234}$ causes repulsion between odd and even parity energy levels. In the consecutive plots in Fig. 2(b), (c) and (d), as we increase $K_{ij}$ from 0.5 to 50, the strength of $K_{ij}$ becomes progressively more dominant over the strength of $J_{1234}$ and the energy level spacings between same parity states become significantly greater than that of different parity states. Similarly, in the bottom set of figures we plot the dependence of energy levels on $K_{ij}$ with $J_{1234}$ fixed in each plot. In Fig. 2(e) we set $J_{1234} = 0$, and observe that states of the same parity split as

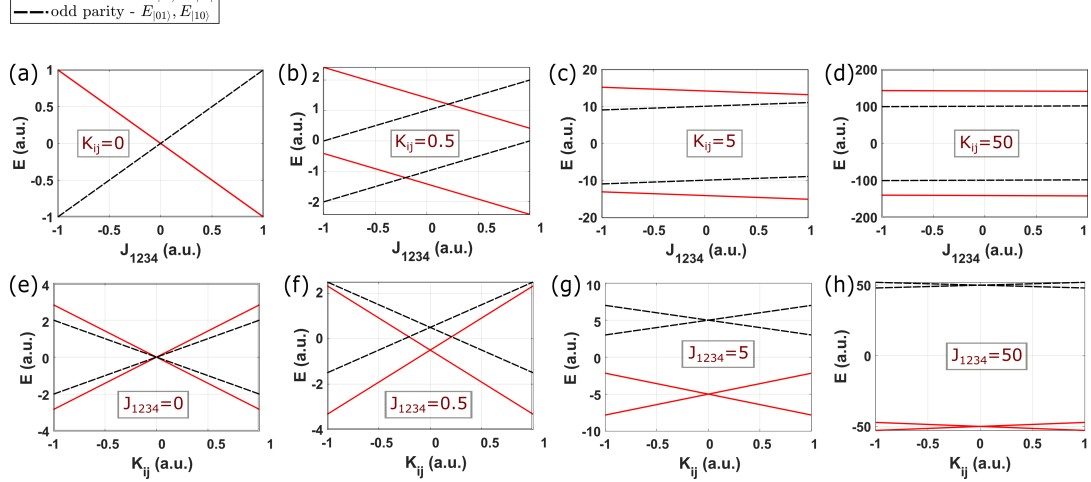

Figure 2: (Top row) Energy levels of the Hamiltonian in Eq. (4) as a function of $J_{1234}$, fixing all $K_{ij}$'s equal to a constant - (a) $K_{ij} = 0$. (b) $K_{ij} = 0.5$. (c) $K_{ij} = 5$. (d) $K_{ij} = 50$. (Bottom row) Energy levels as a function of $K_{ij}$ (all $K_{ij}$'s are equal and varied simultaneously), fixing $J_{1234}$ equal to a constant - (e) $J_{1234} = 0$. (f) $J_{1234} = 0.5$. (g) $J_{1234} = 5$. (f) $J_{1234} = 50$.

function of $K_{ij}$. This shows that $K_{ij}$ causes repulsion between same parity energy levels, and in the consecutive plots in Fig. 2(f), (g) and (h), as we increase $J_{1234}$ from 0.5 to 50 and as the strength of $J_{1234}$ becomes dominant over $K_{ij}$, the energy level spacings between different parity states increase significantly over that of the same parity states. We conclude that $K_{ij}$ causes repulsion between same parity energy levels whereas $J_{1234}$ causes repulsion between different parity energy levels. In the case of the SYK model, since there are only quartic interactions, i.e., only $J_{1234}$ term is non-zero, the energy levels should look like Fig. 2(a), where states of the same parity remain degenerate, while those of opposite parity split, with the energy gap set by $J_{1234}$. Thus, the spectroscopy of the energy differences between the odd and even parity states could tell us which interactions are present in the system.

*Note*: The criteria for energy level crossings in an $N_{cf} > 2$ complex-fermion SYK model might be different than what is shown in Fig. 2(a). For example, for an $N_{cf} = 3$ complex-fermion SYK model, we get energy crossings/degeneracies in opposite parity states rather than the same parity states as in $N_{cf} = 2$ case. This is discussed in Section 6.2 (and shown in Fig. 9(b) and Fig. 10(b)).

## 3   Extracting the interaction strength from tunneling spectroscopy

In this section we discuss the extraction of bilinear and quartic interaction strengths ($K_{ij}$'s and $J_{1234}$ in Eq. (4)) from tunneling transport measurements on hybrid superconductor-semiconductor nanowire devices that host four Majorana zero modes. Generically, transport measurements involve adding or removing electrons from the device and hence these measurements can be used to determine the energy differences between states of different parities, which, in turn, can be used to reconstruct the energy level spectra like those in Fig. 2.

Therefore, our starting point is the spectrum of energy eigenvalues $E_{|00\rangle}$, $E_{|01\rangle}$, $E_{|10\rangle}$, and $E_{|11\rangle}$. The eigenstates are labeled by the occupancy of the two quasiparticle states in the complex fermion representation. For example, the state $|00\rangle$ corresponds to both states being empty, while the state $|01\rangle$ corresponds to the first state empty and the second quasiparticle state occupied. Following the discussion in appendix B, we can relate the parameters of the Hamiltonian in Eq. 5 to the energy eigenvalues via:

$$\begin{pmatrix} E_{|00\rangle} \\ E_{|01\rangle} \\ E_{|10\rangle} \\ E_{|11\rangle} \end{pmatrix} = \begin{pmatrix} 1 & -1 & -1 & -1 \\ 1 & -1 & 1 & 1 \\ 1 & 1 & -1 & 1 \\ 1 & 1 & 1 & -1 \end{pmatrix} \begin{pmatrix} \epsilon_0 \\ \lambda_1 \\ \lambda_2 \\ u \end{pmatrix}, \tag{6}$$

where $\epsilon_0$ is the overall offset of the energy eigenvalues.

Since we are only interested in the energy level differences, identifying each eigenstate by their quasiparticle occupancies becomes redundant. Hence, we refer to each energy level by their parities. From hereon, we will denote $E_{|00\rangle}$ as $E_1^e$, $E_{|10\rangle}$ as $E_1^o$, $E_{|01\rangle}$ as $E_2^o$, and $E_{|11\rangle}$ as $E_2^e$ (where $E^e$'s and $E^o$'s are the quasiparticle energy levels corresponding to the even and odd parity states respectively). The odd and even energy levels can be used interchangeably amongst each other as long as the definition is followed throughout in all the equations.

Inverting the relation in Eq. (6), we can use the spacings between the even and odd parity energy levels to find the Hamiltonian (Eq. (5)) parameters , i.e.,

$$\lambda_1 = (-E_1^e + E_2^e + E_1^o - E_2^o)/4, \tag{7}$$

$$\lambda_2 = (-E_1^e + E_2^e - E_1^o + E_2^o)/4, \tag{8}$$

$$u = (E_1^o + E_2^o - E_1^e - E_2^e)/4. \tag{9}$$

From the derivations in Appendix B, we also show that $u$ gives us a direct relation to the quartic interaction strength $J_{1234}$, but we find that $\lambda$'s do not have a one-to-one relation with $K_{ij}$'s (as seen from Eq. (30)), i.e, a multitude of possible values for $K_{ij}$'s can lead to the same values of $\lambda$'s. However we find that it is possible to set a bound on $K_{ij}$ i.e, make all $K_{ij} = 0$ by setting all $\lambda_i = 0$. This allows us to find an SYK point at which only quartic interactions are present (a non-zero value of $J_{1234}$ and all $K_{ij} = 0$).

From Eq. (7), (8) and (9), we see that $u$ is the difference in energies between the two even parity states and the two odd parity states, whereas $\lambda$'s are made up of energy differences among even states and among odd states. A region with non-zero $u$ but $\lambda_1, \lambda_2 = 0$ will have degenerate $E^e$'s and $E^o$'s with some value of energy gap between them as shown in Fig. 2 (a). Therefore, from these relations we see that characterizing the eigenspectra can help us distinguish between the bilinear and quartic interaction strengths and thus guide us towards an SYK point in an experiment.

## 4  Kitaev chain with interactions

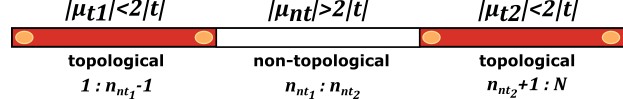

Figure 3: A Kitaev chain nanowire separated into three segments - two topological segments (red) and a non-topological segment (white) in between. The total number of sites is $N$ and the non topological segment ranges between sites $n_{nt_1}$ to $n_{nt_2}$. There are four MZM's (yellow circles) on this wire - two pairs localized at the edges of the two topological segments.

To model an experimentally realizable form of the two complex fermion model, we introduce a 1D Kitaev chain model that hosts four MZMs. Specifically, our model consists of a quantum wire with N sites that is divided into three segments - two topological segments separated by a non-topological segment in the middle, as shown in Fig. 3. In order to induce four-MZM interactions, we supplement the Kitaev chain model with a non-local interaction term that is described in the next section. The total Hamiltonian

$$H_{\text{tot}} = H_{\text{Kitaev-chain}} + H_{\text{nl-int}}, \tag{10}$$

thus consists of the Kitaev chain part $H_{\text{Kitaev-chain}}$ and the non-local interaction part $H_{\text{nl-int}}$. In this section we describe the model. In the next section, we demonstrate that it is possible to tune this model to the point where bilinear interactions become zero while the quartic interaction remains finite.

### 4.1  Kitaev chain

The Hamiltonian of a spinless p-wave Kitaev chain of length $N$ is:

$$
\begin{aligned}
H_{\text{Kitaev-chain}} &= -\sum_j \mu_j c_j^\dagger c_j - t \sum_j c_j^\dagger c_{j+1} + \text{h.c.} \\
&+ \Delta \sum_j c_j c_{j+1} + \text{h.c.},
\end{aligned}
\tag{11}
$$

where $c_j^\dagger$, and $c_j$ are the complex fermion creation and annihilation operators on site $j$. The physical parameters governing this system are the site-dependent electrochemical potential $\mu$, hopping amplitude $t$, and the superconducting pairing field $\Delta$.

As shown in Fig. 3, the first segment of the wire (labeled t1) runs from site 1 to site $n_{\mathrm{nt}_1} - 1$ and is biased to the electrochemical potential $\mu_{\mathrm{t1}}$. The second segment (labeled nt) runs from site $n_{\mathrm{nt}_1}$ to site $n_{\mathrm{nt}_2}$, with electrochemical potential $\mu_{\mathrm{nt}}$. The final segment (labeled t2) runs from site $n_{\mathrm{nt}_2} + 1$ to site $N$ with electrochemical potential $\mu_{\mathrm{t2}}$. The two topological segments, t1 and t2, have their electrochemical potentials set to ensure that they are in the topological phase: $|\mu_{\mathrm{t1}}| < 2|t|$ and $|\mu_{\mathrm{t2}}| < 2|t|$; while the non-topological segment, nt, has its electrochemical potential set to $|\mu_{\mathrm{nt}}| > 2|t|$ to ensure that it is in the trivial phase. MZMs appear at the interface between topological and non-topological segments as indicated in Fig. 3 (here, the vacuum at the ends of the wire can be regarded as trivial).

## 4.2 Four-Majorana coupling from non-local interactions

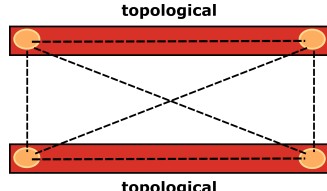

Figure 4: Two Kitaev chain nanowires, each hosting a pair of MZMs at their ends, are placed parallel to each other. This could enhance non-local interactions in the MZMs.

Majorana Zero Modes in the Kitaev chain model are characterized by zero energy states separated from the bulk states by an energy gap. The wave functions corresponding to the Majorana modes have an oscillatory behavior with decaying amplitude. Depending on parameters like the length of the wire ($N$) and other parameters - $\mu$, $t$ and $\Delta$, the MZMs can either be localized at the very ends of the wire (under the condition $t = \Delta$) or spread out into the inner sections of the wire. Interactions between MZMs occur when these wave functions overlap with each other. As the overlap between distant MZMs is typically smaller than between nearby ones, this type of overlap tends to induce bilinear interactions between neighboring MZMs. To introduce a quartic interaction between MZMs, we introduce a non-local interaction term in the Hamiltonian with interaction strength $U$:

$$H_{\mathrm{nl\text{-}int}} = U \sum_{i<j} c_i^\dagger c_i c_j^\dagger c_j . \tag{12}$$

This form of interaction is meant to model long-range interactions mediated, e.g. by charge or by phonons [18, 19]. This interaction could be of Coulomb/density-density origin. It may be possible to enhance the interaction strength by taking certain measures like modifying the device design so as to minimize screening effects from nearby metals. Another idea is to modify the geometry of the device such as instead of having one nanowire with three segments (topo, non-topo, topo), which could have limited long-range interactions, we could have two nanowires (topo segments), with MZMs at their ends, placed parallel to each other, shown in Fig. 4.

In addition to quartic interactions, the non-linear term also induces additional bilinear interactions between the MZMs. In the following section, we show that it is possible to eliminate the bilinear interactions by means of tuning the Hamiltonian parameters.

### 4.3 Connection between the two-complex-fermion model and the Kitaev chain model

The eigenspectrum of the Hamiltonian of the two complex fermion model consists of four energy levels, that are separable in parity sectors as the total Hamiltonian is parity preserving. Tuning the interaction strengths relative to one another affect the energy level spacings specified by their parities as shown in Eqs. (7), (8), (9).

In the Kitaev chain model, the energy levels corresponding to the MZMs are the lowest four energy levels separated from the bulk states by an energy gap. The energy level structure of these four lowest energy levels is dependent on the Hamiltonian parameters. By applying equations (7), (8), (9) to these four lowest energy levels we can construct an effective low-energy model and extract its interaction parameters in terms of the MZM representation of the Hamiltonian Eq. (4).

## 5 Reducing bilinear interactions while enhancing the quartic interaction strength

Once we are able to identify the interactions between the MZMs, our goal becomes tuning the system to an SYK point at which the bilinear interactions become zero while the quartic interaction remains finite. In order to achieve this goal we tune the Hamiltonian parameters of our system (defined by $\mu_{t1}$, $\mu_{t2}$, $\mu_{nt}$, $t$, $\Delta$ and $U$) in order to zero out the bilinear interactions, i.e. $|\lambda_1|, |\lambda_2| \simeq 0$, while at the same time maximizing the value of quartic interaction $|u|$.

In this section, we first sweep the system parameters and show the existence of multiple, approximate SYK points. Next, we use an advanced optimization algorithm (the hybrid Genetic Algorithm described below) to locate SYK points at which bilinear interactions become essentially zero.

### 5.1 Existence of approximate SYK points

To find approximate SYK points, we begin with the 1D Kitaev chain model of Eq. (11) with no long-range interactions. When the wave functions of the MZMs overlap, the associated energy levels become non-zero. However, as the MZM wave functions are oscillatory, it is possible to tune the overlap integral to be zero in certain parametric regime/points where the overlapping wave functions cancel each other out [20]. We search for such optimal points where the lowest two quasiparticle energies are close to zero (i.e. the lowest four many-body states are degenerate) despite the MZM wave function envelopes having significant overlap. An example for this has been shown in Appendix D where we plot MZM wave functions and show how they overlap at an optimal point (shown in Fig. 13). After finding one of these optimal points, we add the non-local interaction term and show that by sweeping across a range of non-local interaction strength $U$, we can find approximate SYK points.

Therefore, we optimize for the condition $|\lambda_1|, |\lambda_2| = 0$ for the Hamiltonian in Eq. (10) at $U = 0$, i.e., without the presence of the non-local interaction which suppresses all nearest neighbor Majorana interactions. We then show that if we add the non-local interaction term $U$ at these special optimized points, we can effectively have a dominant quartic interaction strength.

#### 5.1.1 Obtaining the optimal points

We show in the Appendices that there are points in the parameter space of $\mu_{t1}$, $\mu_{t2}$, $\mu_{nt}$, $t$ and $\Delta$ at which we can minimize hybridization energy of the overlapping MZMs, i.e, tune

their overlap integral to zero. We discuss the role of these parameters for optimization in Appendix C (see, Fig. 12 for a summary). At these optimal points the lowest four many-body energy levels are close to degenerate. Hence, we perform a numerical search for points where $\left|E_{|11\rangle} - E_{|00\rangle}\right|$ is minimized. The values for $\mu_{t1}$, $\mu_{t2}$, $\mu_{nt}$, $t$ and $\Delta$ are obtained using a global search algorithm which we discuss in details in Appendix E. The optimal parameters obtained for $N = 10$ complex fermions and non-topological region - $n_{nt_1} = 4$, $n_{nt_2} = 7$ are: $t = 0.4025$, $\Delta = 0.2167$, $\mu_{t1} = 0.4832$, $\mu_{t2} = 0.4832$, $\mu_{nt} = 8.5364$.

### 5.1.2 Sweeping interactions to find approximate SYK points

Having minimized the local bilinear interactions by tuning the system to an optimal point in the parameter space of $\{\mu_{t1}, \mu_{t2}, t, \Delta\}$, we then add the non-local interaction term $H_{\text{nl-int}}$ to induce quartic interactions between the MZMs. However, this non-local interaction term can also introduce additional non-local bilinear interactions besides the quartic interaction. Thus, we further optimize to cancel out these additional bilinear terms by tuning the non-local interaction strength $U$ to the SYK point, i.e., we sweep across a range of values for $U$ to find a point where $|\lambda_1|, |\lambda_2| \simeq 0$ and $|u|$ is at a maximum value. This is shown in Fig. 5(a), where we fix the values of $\{\mu_{t1}, \mu_{t2}, t, \Delta\}$ and sweep the value of $U$. In this figure we see that $\lambda_1$ (solid yellow line), $\lambda_2$ (dashed blue line) and $u$ (solid purple line) have several local maxima and minima at various points of $U$. Both $\lambda_1$ and $\lambda_2$ follow the same trajectory, i.e, their extrema coincide, and $u$, having a different trajectory, has extrema at different points. At $U = 0.0733$ and $U = 0.2118$, $\lambda$'s have minima and $u$ has maxima, i.e, they are the SYK points (shown by vertical dotted line-cuts). The best SYK point characterized by the greatest value for $\left|\frac{u}{\lambda}\right|$ has been shown in this figure with a dotted black line-cut at $U = 0.212$ at which we get $\lambda_1, \lambda_2 \simeq 8.28e^{-5}$ and $u \simeq 0.036$ ($\left|\frac{u}{\lambda}\right| \simeq 435$).

In Fig. 5(b), we plot the energies of the lowest four states, which were used to construct Fig. 5(a), as a function of $U$. We see that at the best SYK point (shown with the black dotted line-cut, zoomed-in in the inset) the even parity (red lines) and odd parity (solid black lines) energy levels are degenerate, i.e, $\left|E_{|11\rangle} - E_{|00\rangle}\right| \simeq 0$, $\left|E_{|10\rangle} - E_{|01\rangle}\right| \simeq 0$), with a finite energy gap between them, i.e. $\left|E_{|11\rangle} + E_{|00\rangle}\right| - \left|E_{|10\rangle} + E_{|01\rangle}\right| \neq 0$. Following the discussions in Section 2.1, this level structure is similar to the one displayed in Fig. 2 (a) with $K_{ij} = 0$ and $J_{1234} \neq 0$. Thus, from the low energy level spectral point of view, this point matches our expectation for an SYK point.

Other than the non-local interaction term Eq. (12), we have also explored other interaction terms but could not find an SYK point as we swept through a range of the interaction strength $U$ at the optimal point. This is discussed in Appendix G and some examples are shown in Fig. 16). We hypothesize that our inability to tune systems with alternative interactions to an SYK point is due to the more local nature of the alternative interactions.

*Note*: There is a constraint on the feasible range of $U$ that we have access to in the optimization process. This is based on the condition that MZM's appear in the topological regime separated by an energy gap from the bulk states ($\simeq \Delta$). Specifically, varying $U$ beyond a certain bound results in the closing of this energy gap and the penetration of the MZM states into the continuum of bulk quasiparticle states. In order to detect this possibility, as we tune $U$, we check the total parity of the four lowest energy (many-body) states. The total parity is expected to be zero as there are two even and two odd parity states. However as the MZM states penetrate into the continuum, the parity of the lowest energy states becomes random and we know that we have exceeded the valid range of $U$.

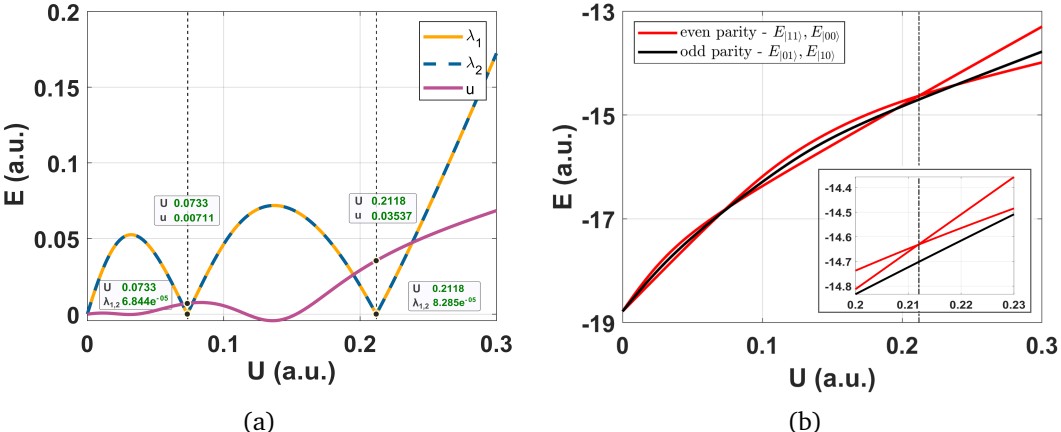

(a)                 (b)

Figure 5: The optimal point parameters obtained for a $N = 10$ site Kitaev chain model (non-topological segment: $n_{\text{nt}_1} = 4, n_{\text{nt}_2} = 7$) obtained following the method of finding approximate SYK point by minimizing MZM wave function overlap to zero as described in Section 5.1 are: $t = 0.4025$, $\Delta = 0.2167$, $\mu_{\text{t}1} = 0.4832$, $\mu_{\text{t}2} = 0.4832$, $\mu_{\text{nt}} = 8.5364$. (a), We sweep across $U$ at the optimal point and find a couple of SYK points at $U = 0.0733$ and $U = 0.2118$. At the best SYK point, i.e. at $U = 0.2118$, we have $\lambda_1, \lambda_2 \simeq 8.28e^{-5}$ and $u \simeq 0.036$. (b), We plot the lowest four energy levels vs $U$ labelled by their parities (even-red, odd-black). At the best SYK point obtained at $U = 0.2118$ (shown in inset), that the two even and the two odd parity states are degenerate (level-crossings) while the energy gap between between different parity states is at a maximum.

## 5.2 Using a search algorithm to find SYK points

Another way to look for SYK points is to implement a direct search for $|\lambda_1|, |\lambda_2| \simeq 0$ and a maximum $|u|$, through the entire parameter space. We intend to not only look for better optimized points but also to compare with the results of the previous subsection in which we tuned for zero MZM wave function overlap integral.

Our optimization problem now consists of a ($|\mu_{\text{t}1}| < 2|t|$, $|\mu_{\text{t}2}| < 2|t|$, $|\mu_{\text{nt}}| > 2|t|$) and constraints in objective function. Hence, we attempt to solve this problem using an advanced optimization technique - a hybrid Genetic algorithm (a Genetic Algorithm search for roughly locating the SYK points, followed by a derivative-based search for refining the location of the SYK points). Our objective is to maximize $|u|$ under the constraints of (i) $|\lambda_1|, |\lambda_2| \simeq 0$, and (ii) total parity of lowest four eigenstates equals zero. This search yields optimized results for all the parameters $\mu_{\text{t}1}$, $\mu_{\text{t}2}$, $\mu_{\text{nt}}$, $t$, $\Delta$ and $U$ corresponding to the best SYK point in the given search range (discussed in details in Appendix F). For $N = 10$ complex fermions with non-topological region - $n_{\text{nt}_1} = 4$, $n_{\text{nt}_2} = 7$, the values obtained are $t = 0.3229$, $\Delta = 0.1$, $\mu_{\text{t}1} = 0.5871$, $\mu_{\text{t}2} = 0.5871$, $\mu_{\text{nt}} = 6.3944$ and $U = 0.2254$.

In Fig. 6(a), we plot $\lambda_1$, $\lambda_2$, and $u$ as a function $U$ to show a comparison with the results in Fig. 5(a) (from the previous method). We fix the values of $\{\mu_{\text{t}1}, \mu_{\text{t}2}, t, \Delta\}$ and sweep the value of $U$ so as to intersect the SYK point found by the hybrid Genetic Algorithm at $U = 0.2254$. We see that $\lambda$'s and $u$ follow similar trajectory in both the figures with the extrema of $\lambda$'s located at different points than from that of $u$, and we can indeed find three SYK points at $U = 0.0176, 0.0891$, and $0.2254$. The best SYK point can be seen at $U = 0.2254$ (shown by black dotted line cut) where $\left|\frac{u}{\lambda}\right| \simeq 530$. Likewise, in Fig. 6(b), we show that at the SYK point $\left|E_{|11\rangle} - E_{|00\rangle}\right| \simeq 0$, $\left|E_{|10\rangle} - E_{|01\rangle}\right| \simeq 0$ while $\left|E_{|11\rangle} + E_{|00\rangle}\right| - \left|E_{|10\rangle} + E_{|01\rangle}\right| \neq 0$, similar to that in Fig. 5(b). We note that the SYK point obtained by this method (Fig. 6) yields a better result

(higher value for $\left|\frac{u}{\lambda}\right|$) than the previous case (as shown in Fig. 5).

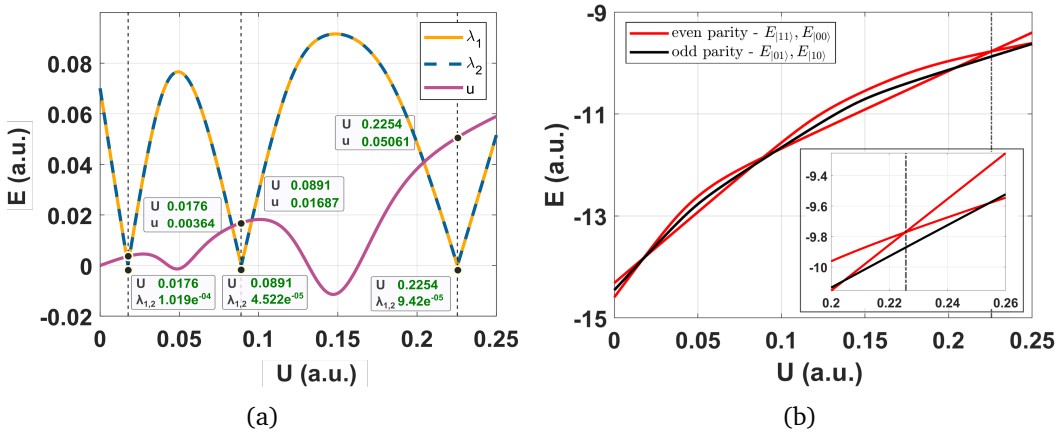

Figure 6: The SYK point parameters obtained for a $N = 10$ site fermion Kitaev chain model (non-topological segment: $n_{nt_1} = 4, n_{nt_2} = 7$) obtained using a hybrid genetic algorithm search as described in Section 5.2 are: $t = 0.3229$, $\Delta = 0.1$, $\mu_{t1} = 0.5871$, $\mu_{t2} = 0.5871$, $\mu_{nt} = 6.3944$, $U = 0.2254$. (a), We sweep across $U$ fixing the other parameters so as to meet the best SYK point at $U = 0.2254$. We find three SYK points at $U = 0.0176$, $U = 0.0891$ and $U = 0.2254$. At the best SYK point, i.e. at $U = 0.2254$, we have $\lambda_1, \lambda_2 \simeq 9.42e^{-5}$ and $u \simeq 0.05$. (b), We plot the lowest four energy levels vs $U$ labelled by their parities (even-red, odd-black). At the SYK point obtained (shown in inset), we find that the two even and the two odd parity states are degenerate (level-crossings) while the energy gap between between different parity states is at a maximum.

### 5.2.1 Adding another parameter for optimization

In the four-Majorana Hamiltonian (Eq. (4)) we have seven independent terms: six bilinear terms ($K_{ij}$) and one quartic term ($J_{1234}$). In the Kitaev chain Hamiltonian, it is reasonable then to expect that we find a better optimal point by expanding our parameter space to seven by adding another independent variable. For this we have set the tunneling amplitude parameter $t_c$ at the center of the wire different from $t_e$ at the edges as two independent parameters. Then, we implemented the same hybrid Genetic Algorithm to find SYK points, and plot the results in Fig. 7. In Fig. 7(a) we see that $\lambda$'s and $u$ follow quite a different trajectory than in the previous figures (Fig. 5(a), Fig. 6(a)) but we are still able to find an SYK point at $U = 0.3477$ where $\left|\frac{u}{\lambda}\right| \approx 60,000$. By this measure, adding a seventh parameter gives a better SYK point than the previous two cases. Similar to the previous cases, by plotting the unprocessed energy levels in Fig. 7(b) we observe that at $U = 0.3477$, $\left|E_{|11\rangle} - E_{|00\rangle}\right| \simeq 0$, $\left|E_{|10\rangle} - E_{|01\rangle}\right| \simeq 0$ while $\left|E_{|11\rangle} + E_{|00\rangle}\right| - \left|E_{|10\rangle} + E_{|01\rangle}\right| \neq 0$.

## 6 Extension to a six MZM model

### 6.1 Why do we extend the model?

In the sections above we deal with characterizing and enhancing the quartic interaction strength within four-Majorana models. However, in the full SYK model, we need to have a large numer of Majorana zero modes with multiple quartic interaction terms. Hence, to characterize and measure multiple quartic interactions, we need to extend our model from $N_\gamma = 4$

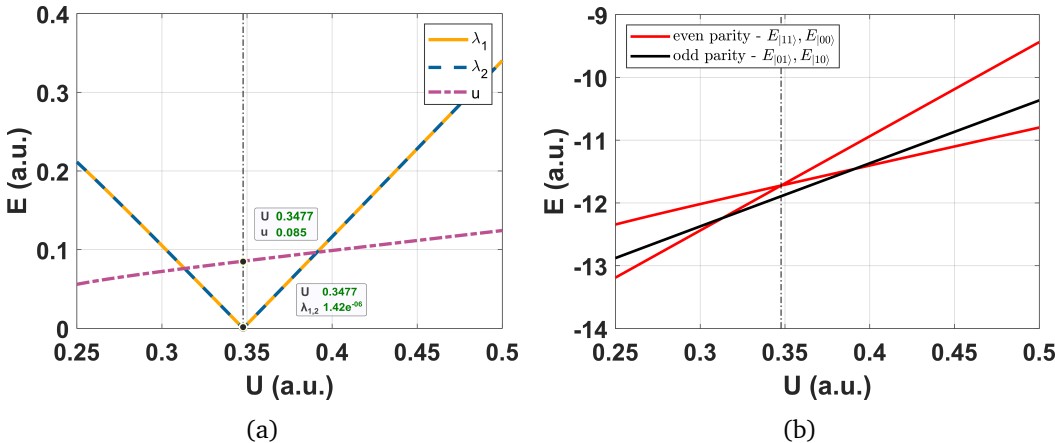

Figure 7: We add another parameter (tunneling amplitude - $t_c$ at the center of the wire different than at the edges - $t_e$) for optimization in order to find better SYK points as described in Section 5.2.1. The SYK point parameters for $N = 10$ site Kitaev chain model (non-topological segment: $n_{nt_1} = 4, n_{nt_2} = 7$) obtained using a hybrid genetic algorithm search are: $t_e = 0.9028$, $t_c = 0.1166$, $\Delta = 0.1041$, $\mu_{t1} = 0.3299$, $\mu_{t2} = 0.3299$, $\mu_{nt} = 7.2691$ and $U = 0.3477$. (a), We sweep across $U$ fixing the other parameters to meet the SYK point at $U = 0.3477$. At this point, we have $\lambda_1, \lambda_2 \simeq 1.42e^{-6}$ and $u \simeq 0.085$. (b), We plot lowest four energy levels vs $U$ labelled by their parities (even-red, odd-black). At the SYK point obtained (shown in inset), we find that the two even and the two odd parity states are degenerate (level-crossings) while the energy gap between between different parity states is at a maximum.

to higher $N_\gamma$. As a proof of concept of our model being extendable to higher $N_\gamma$, we try applying the methods discussed in the sections above to a six-Majorana model.

| $\|\mu_{t1}\|<2\|t\|$ | $\|\mu_{nt1}\|>2\|t\|$ | $\|\mu_{t2}\|<2\|t\|$ | $\|\mu_{nt2}\|>2\|t\|$ | $\|\mu_{t3}\|<2\|t\|$ |
|---|---|---|---|---|
| topological | non-topological | topological | non-topological | topological |
| $1 : n_{nt1_1}$-1 | $n_{nt1_1} : n_{nt1_2}$ | $n_{nt1_2}$+1 : $n_{nt2_2}$-1 | $n_{nt2_1} : n_{nt2_2}$ | $n_{nt2_2}$+1 : N |

Figure 8: A Kitaev chain nanowire separated into five segments - three topological segments (red) and two non-topological segment (white). The total number of sites is $N$ and the non topological segments range between sites $n_{nt1_1}$ to $n_{nt1_2}$ for non-topological segment 1, and $n_{nt2_1}$ to $n_{nt2_2}$ for non-topological segment 2. There are six MZM's (yellow circles) on this wire - three pairs localized at the edges of the three topological segments.

## 6.2 The three-complex-fermion model

This model consists of three sites, each with two Majoranas, i.e. six Majoranas in total. There can be three kinds of interaction terms between six Majoranas, they are - (i) 15 bilinear interaction terms, (ii) 15 quartic interaction terms and (iii) 1 sextic interaction term - which is the new term that appears with six MZMs. In the MZM representation, the Hamiltonian of this model is:

$$H = J_{123456} \prod_{i=1}^{6} \gamma_i + \sum_{1 \le i < j < k < l \le 6} J_{ijkl} \gamma_i \gamma_j \gamma_k \gamma_l + i \sum_{1 \le i < j \le 6} K_{ij} \gamma_i \gamma_j, \tag{13}$$

where $J_{123456}$ is the sextic interaction strength, $J_{ijkl}$'s are the quartic interaction strengths and $K_{ij}$'s are the bilinear interaction strengths.

## 6.3 Extracting the interaction strengths

When we diagonalize the Hamiltonian in Eq. 13, we get eight eigenvalues corresponding to the $2^3 \times 2^3$ Hilbert space. The eigenstates are in the form $|n_1 n_2 n_3\rangle$ where $n_1, n_2, n_3 = 0$ or 1 for empty or filled fermion quasiparticle states respectively. As an extension to the four Majorana model, the Hamiltonian can be written in the quasiparticle basis as:

$$
\begin{aligned}
H \;=\;& \epsilon_0 + \lambda_1(2n_1-1) + \lambda_2(2n_2-1) + \lambda_3(2n_3-1) \\
+\;& u_{12}(2n_1-1)(2n_2-1) + u_{13}(2n_1-1)(2n_3-1) \\
+\;& u_{23}(2n_2-1)(2n_3-1) \\
+\;& v(2n_1-1)(2n_2-1)(2n_3-1),
\end{aligned}
\tag{14}
$$

where $\lambda$'s are the bilinear interaction strengths, $u$'s are the quartic interaction strengths and $v$ is the sextic interaction strength in the quasiparticle basis.

Following the arguments as in Appendix B and Section 3, we can conclude that in a similar manner for the six MZMs case the energy levels of states $|n_1 n_2 n_3\rangle$ are related to interaction strengths $\lambda$'s, $u$'s and $v$ as:

$$
E = AI,
\tag{15}
$$

where $E$ is the column matrix of energy levels for states $|n_1 n_2 n_3\rangle$, where $n_1, n_2, n_3 = 0, 1$ in the sequence as shown below:

$$
E = \begin{pmatrix} E_{|000\rangle} \\ E_{|001\rangle} \\ E_{|010\rangle} \\ E_{|100\rangle} \\ E_{|011\rangle} \\ E_{|101\rangle} \\ E_{|110\rangle} \\ E_{|111\rangle} \end{pmatrix},
\tag{16}
$$

where $I$ is the column matrix of interaction strengths $\lambda$'s (bilinear), $u$'s (quartic) and $v$ (sextic), i.e.

$$
I = \begin{pmatrix} \epsilon_0 \\ \lambda_1 \\ \lambda_2 \\ \lambda_3 \\ u_{12} \\ u_{13} \\ u_{23} \\ v \end{pmatrix},
\tag{17}
$$

and A is the matrix transforming $E$ to $I$ :

$$
A = \begin{pmatrix}
1 & -1 & -1 & -1 & 1 & 1 & 1 & -1 \\
1 & -1 & -1 & 1 & 1 & -1 & -1 & 1 \\
1 & -1 & 1 & -1 & -1 & 1 & -1 & 1 \\
1 & 1 & -1 & -1 & -1 & -1 & 1 & 1 \\
1 & -1 & 1 & 1 & -1 & -1 & 1 & -1 \\
1 & 1 & -1 & 1 & -1 & 1 & -1 & -1 \\
1 & 1 & 1 & -1 & 1 & -1 & -1 & -1 \\
1 & 1 & 1 & 1 & 1 & 1 & 1 & 1
\end{pmatrix}.
\tag{18}
$$

Thus we can extract the interaction strengths by inverting this relation to solve for $I$:

$$I = A^{-1}E. \tag{19}$$

## 6.4 Kitaev chain model with interactions

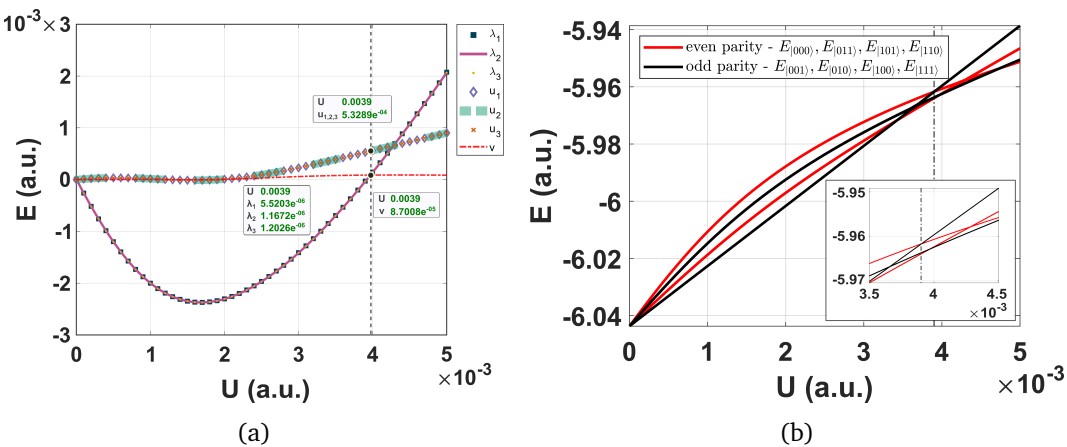

| | |
|---|---|
| (a) | (b) |

Figure 9: The SYK point parameters for three-complex-fermion model for a $N = 10$ site Kitaev chain model (first non-topological segment: $n_{\mathrm{nt1}_1} = 3$, $n_{\mathrm{nt1}_2} = 4$, second non-topological segment: $n_{\mathrm{nt2}_1} = 7$, $n_{\mathrm{nt2}_2} = 8$) obtained using the approximate method for SYK point search, i.e., method (i) described in Section 6.5 are: $t = 0.0144$, $\Delta = 0.0104$, $\mu_{\mathrm{t1}} = 0.0100$, $\mu_{\mathrm{t2}} = 0.0100$, $\mu_{\mathrm{t3}} = 0.0100$, $\mu_{\mathrm{nt1}} = 3.0001$, $\mu_{\mathrm{nt2}} = 3.0001$, $U = 0.0039$. (a), At the SYK point (shown by black dotted line-cut) $\lambda_1 = 5.5203e^{-06}$, $\lambda_2 = 1.1672e^{-06}$, $\lambda_3 = 1.2026e^{-06}$, $u_{12} = u_{13} = u_{23} = 5.3289e^{-04}$ and $v = 8.7008e^{-05}$ (b), At the SYK point, there is degeneracy between one even and one odd parity level. Focusing on the line-cut in the inset, the bottom most line (red) consists of three degenerate even parity energy levels and the second line from bottom (black) consists of three degenerate odd parity energy levels, the two intersecting lines on the top are a single odd and a single even parity energy level each.

Our goal is to model a Kitaev chain quantum wire (similar to the four MZMs case) that generates six MZMs. It is an $N$ site chain that we divide into five segments - three topological segments separated by two non-topological segments in between two topological segments (topo-nontopo-topo-nontopo-topo). Following the conditions for topological phase we have $|\mu_{\mathrm{t1}}| < 2|t|$, $|\mu_{\mathrm{t2}}| < 2|t|$ and $|\mu_{\mathrm{t3}}| < 2|t|$ for the topological segments t1, t2, t3, and $|\mu_{\mathrm{nt1}}| > 2|t|$, $|\mu_{\mathrm{nt2}}| > 2|t|$ for the non-topological segments nt1, nt2. As a result, we get six MZMs at the ends of the three topological segments. This has been shown schematically in Fig. 8.

To introduce the quartic interactions in the MZMs, we add the same non-local interaction term as we did in the four MZM case which is:

$$H_{\mathrm{nl\text{-}int}} = U \sum_{i<j} c_i^\dagger c_i c_j^\dagger c_j. \tag{20}$$

The total Hamiltonian after adding up the interaction term is:

$$H_{\mathrm{tot}} = H_{\mathrm{Kitaev\text{-}chain}} + H_{\mathrm{nl\text{-}int}}. \tag{21}$$

This also introduces additional non-local bilinear and sextic interactions in between the MZMs which we can suppress as shown in the following sections.

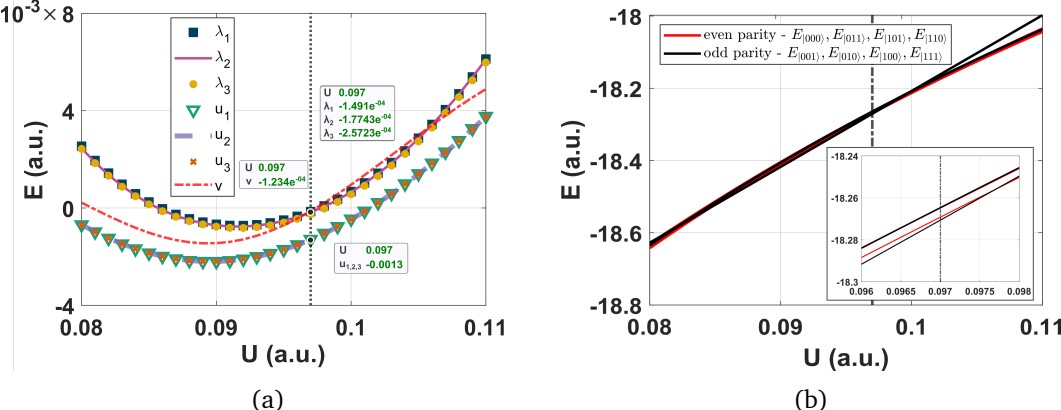

Figure 10: The SYK point parameters for three-complex-fermion model for an $N = 10$ site Kitaev chain model (first non-topological segment: $n_{nt1_1} = 3$, $n_{nt1_2} = 4$, second non-topological segment: $n_{nt2_1} = 7$, $n_{nt2_2} = 8$) obtained using a hybrid genetic algorithm search, i.e., method (ii) described in Section 6.5 are: $t = 0.1$, $\Delta = 0.1315$, $\mu_{t1} = 0.5746$, $\mu_{t2} = 0.5743$, $\mu_{t3} = 0.5746$, $\mu_{nt1} = 10$, $\mu_{nt2} = 10$, $U = 0.097$. (a) At the SYK point at $U = 0.097$ (shown by black dotted line-cut) $\lambda_1, = -1.4910e^{-04}$, $\lambda_2 = -1.7743e^{-04}$, $\lambda_3 = -2.5723e^{-04}$, $u_{12} = u_{13} = u_{23} = -0.0013$ and $v = -1.234e^{-04}$. (b) At the SYK point, there is degeneracy between one even and one odd parity level. Focusing on the line-cut in the inset, the bottom two red and black lines are single levels whereas the top line consists of three degenerate even (red) and odd parity levels (black) crossing each other.

## 6.5 Search for SYK points for dominant quartic interaction strength and suppressing bilinear and sextic interaction strengths

Following the same approach as in Section 5, we enhance the quartic interaction strength using two different methods. (i) By optimizing the MZM wave function overlap integral to zero, i.e., searching for global minima for $\left|E_{|111\rangle} - E_{|000\rangle}\right|$ in the parameter space of $\{\mu_{t1}, \mu_{t2}, \mu_{t3}, \mu_{nt1}, \mu_{nt2}, t, \Delta\}$. Then by sweeping the value of $U$, we search for minima of bilinear and sextic interaction strengths and maxima of quartic interaction strengths, i.e., maximize $|u_{12}|, |u_{13}|, |u_{23}|$ and set $|\lambda_1|, |\lambda_2|, |\lambda_3|, |v| \simeq 0$, within a feasible search range such that the energy gap between the low energy states and bulk states is maintained. (ii) By performing a direct search for SYK points within the total parameter space of $\mu_{t1}, \mu_{t2}, \mu_{t3}, \mu_{nt1}, \mu_{nt2}, t, \Delta$ and $U$. We use a hybrid Genetic Algorithm search to optimize for $|\lambda_1|, |\lambda_2|, |\lambda_3|, v \simeq 0$ and maximize $|u_1|, |u_2|, |u_3|$ with the constraint that the energy gap between the MZM states and bulk states doesn't close.

Using method (i) we suppress all local bilinear interactions by finding optimal points where MZM wave function overlap integral is zero. We find optimal points where $\left|E_{|111\rangle} - E_{|000\rangle}\right| \simeq 0$ using a global search algorithm similar to that in four MZM case (described in details in Appendix E.1), then upon sweeping through $U$, we can find SYK points as shown in Fig. 9 where all interactions except for the quartic interactions are well suppressed. In Fig. 9(a), following the black dotted line-cut, at the SYK point we have $\left|\frac{u}{\overline{\lambda}}\right| \simeq 200$ ($\overline{\lambda}$: average of $\lambda_{1,2,3}$, $u = u_{12} = u_{13} = u_{13}$) and $\left|\frac{u}{v}\right| \simeq 6$.

The energy levels used to compute the parameters $\lambda_{1,2,3}$, $u_{12,23,13}$, and $v$ in Fig. 9(a) are plotted as a function of $U$ in Fig. 9(b). In this figure, there are eight energy levels with some degenerate levels. The black dotted line-cut corresponds to the SYK point. Focusing on the line-cut in the inset, the bottom most line (red) consists of three degenerate even parity energy

levels and the second line from bottom (black) consists of three degenerate odd parity energy levels, the two intersecting lines on the top are a single odd and a single even parity energy level each. Thus, at the SYK point we observe energy crossings between even and odd parity levels, in contrast to the four MZM case, in which states of the same parity were crossing.

Using method (ii) we find more SYK points using a hybrid Genetic Algorithm (described in detail in Appendix F.1). In Fig. 10 we plot the interaction strengths as a function of $U$ to capture the SYK point found by the Genetic Algorithm at $U = 0.097$. Here, $\left|\frac{u}{\lambda}\right| \simeq 2$ ($\overline{\lambda}$: average of $\lambda_{1,2,3}$, $u = u_{12} = u_{13} = u_{13}$) and $\left|\frac{u}{v}\right| \simeq 10.5$. For completeness, we plot the energy levels that we used to extract $\lambda_{1,2,3}$, $u_{12,23,13}$, and $v$ (in Fig. 10(a)) in Fig. 10(b). The level structure is analogous to the one previously found in Fig. 9(b). In the inset, the bottom two red and black lines are single levels whereas the top line consists of three degenerate even (red) and odd parity levels (black) crossing each other.

We note that the eigenstates are shuffled at the energy level crossings and we need to reorder them in order to maintain consistency of the definitions of $\lambda$'s, $u$'s and $v$ at different search intervals/points. This was done using an eigenvalues reordering algorithm [21].

# 7 Future Relevance

From this study we show that it is indeed possible to design a low $N_\gamma$ (particularly $N_\gamma = 4, 6$) SYK model in a 1D nanowire system. It might be possible to extend this model to a $N_\gamma > 6$ system in future work. We show that it is possible to analyze the experimentally accessible spectra of eigenstates with quantum transport measurements, and use this information to assess the strength of bilinear and quartic interaction terms.

In hybrid superconductor-semiconductor nanowire devices, some of the parameters that we use in constructing the multi-segment Kitaev chain model are tunable. For instance, the tunneling amplitudes and chemical potentials can be tuned with gates. Other parameters, such as the induced gap, may be harder to tune in situ, though it is possible in principle. The biggest anticipated challenge is to tune the interaction strength $U$ which may be severely constrained by the device geometry. This crucial term may also turn out to be too small, thus closing the door to future work. Though it can possibly be enhanced by careful design of nanowire devices.

# 8 Experimental Protocol for a 4-MZM interaction device

We propose a semiconductor-superconductor nanowire device with tunnel probes connected to the ends of the wire acting as source and drain channels across which a voltage bias is applied for performing tunneling spectroscopy. This device can be fabricated to have multiple local gates in contact/close to the nanowire that can be individually controlled to tune the system parameters such as $\mu$ and $t$ corresponding to different segments of the wire. Other parameters like $\Delta$ can be pre-selected by choosing appropriate material for the superconductor-semiconductor nanowire. We can vary $U$ by playing with the geometry of the device, e.g., by having two different nanowires, that host MZMs, placed parallel to each other, by minimizing screening from nearby metals and by tuning the electron density.

Differential conductance data will tell us the energy level spacings and thus can be used to construct the lowest four quasiparticle eigenspectra similar to Fig. 2. Then, by using Eq. (7), (8) and (9) we can extract the values of $\lambda_1$, $\lambda_2$ and $u$ for a particular set of conditions. Varying the system parameters will give several sets of spectroscopy data for different conditions, which

can be further analyzed to tune the system to SYK point, i.e., zero $\lambda$'s and a maximum $u$. We can vary a couple of parameters at a time to obtain a map of energy level spacings as shown in Fig. 12 , which will give us an estimate of the parametric space for the approximate SYK points. Then we can further narrow down our search by tuning individual parameters and obtaining the interaction strengths as a function of these parameters, as shown in Fig. 5(a), 6(a), 7(a) where we plot $\lambda$'s and $u$ as a function of $U$. Alternatively, we can fix $U$ and obtain the interaction strengths as a function of other parameters as shown in Fig. 16. From this, we can also conclude that it is not necessary to tune all the parameters simultaneously to arrive at the SYK point once we have a good parametric region to work with.

# 9 Further Reading

For further background on topological states of matter and Majorana zero modes we recommend these papers. [22–25, 25–28]. Variants of the SYK model relevant to condensed matter physics: [29–31].

Discussions of the SYK Theory can be found here [1–4, 32–36].

Works on the OTOC and Lyapunov Exponent: [5, 6].

Theories that include interacting MZMs: [18, 19, 37–42].

Proposed experimental models and simulations of SYK model: [8–17, 43].

Quantum transport studies related to the SYK model: [44–47].

# 10 Study Design

This study was undertaken in a period of about nine months. A summary of our study design is shown in Fig. 11.

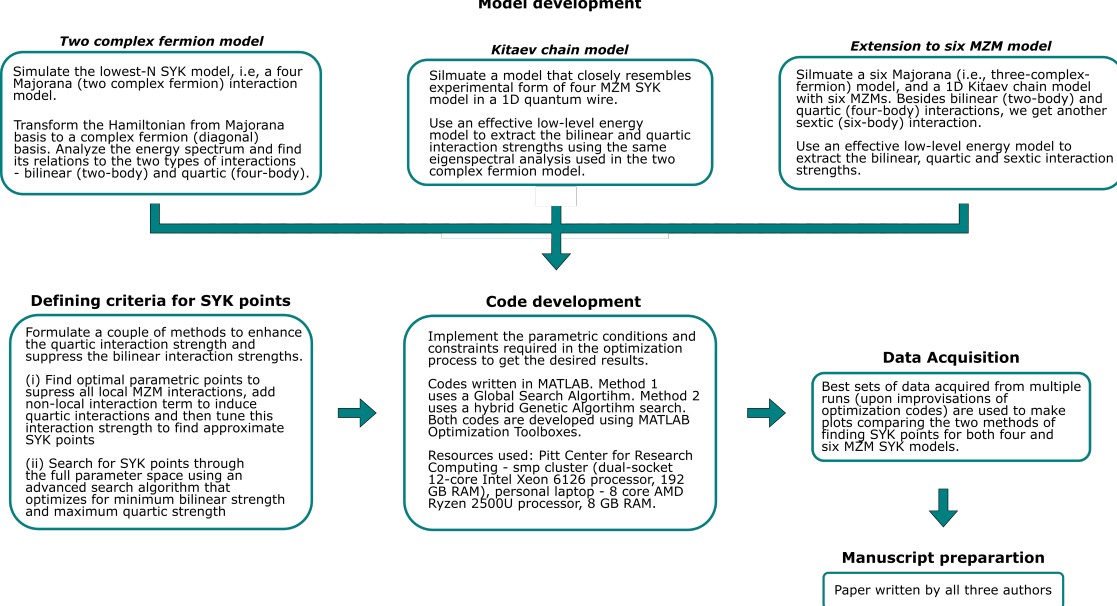

Figure 11: Study design diagram.

## Data Availability

All codes are available on Zenodo [48].

## Acknowledgements

The authors thank J. Stenger for discussions of the SYK model.

### Funding

S.F. and D.P. are supported by NSF PIRE-1743717. S.F. is supported by NSF DMR-1906325, ONR and ARO.

## A   Constructing gamma matrices

The matrices representing the $\gamma$ operators are constructed by transforming the Hamiltonian into spin chain basis (similar to Jordan Wigner transformation) using the following algorithm.

Based on the Clifford algebra,

$$\{\gamma_\mu, \gamma_\nu\} = \delta_{\mu\nu} I_{2^{N_\gamma/2} \times 2^{N_\gamma/2}}; \qquad \mu, \nu = 1....N_\gamma. \tag{22}$$

For any $N_\gamma$, we can build the matrix representation of $\gamma_\mu$ by taking product of Pauli matrices. Pauli matrices ($\sigma_i$'s) satisfy the Clifford algebra and hence forms the representation:

$$\{\sigma_i, \sigma_j\} = \delta_{ij} I_{2\times 2}. \tag{23}$$

So starting with $N_\gamma = 2$, we can take $\gamma_1 = \sigma_2$, and $\gamma_2 = \sigma_1$. We can use an iterative approach to obtain the matrices for a higher dimension $N_\gamma$. Let $\gamma_\mu(\mu = 1....N_\gamma - 2)$ be a $2^{N_\gamma/2-1} \times 2^{N_\gamma/2-1}$ matrices. Then $2^{N_\gamma/2} \times 2^{N_\gamma/2}$ matrices $\gamma_{\widetilde{\mu}}(\widetilde{\mu} = 1...N_\gamma)$ are given as:

$$\gamma_{\widetilde{\mu}} = \gamma_\mu \otimes -\sigma_3, \qquad \text{for } \widetilde{\mu} = 1,...,N_\gamma - 2, \tag{24}$$

$$\gamma_{N_\gamma-1} = I_{2^{N_\gamma/2} \times 2^{N_\gamma/2}} \otimes \sigma_2, \ \ \gamma_{N_\gamma} = I_{2^{N_\gamma/2} \times 2^{N_\gamma/2}} \otimes \sigma_1. \tag{25}$$

## B   Connecting the four MZM and the two complex fermion representations

From the two-complex-fermion model [section 2], a Hamiltonian with interacting four MZM can be represented as

$$H = H_{\text{bl}} + H_{1234} = i \sum_{1 \leq i < j \leq 4} K_{ij} \gamma_i \gamma_j + J_{1234} \prod_{i=1}^{4} \gamma_i. \tag{26}$$

Here $\gamma$'s represent the four MZM's, $K_{ij}$'s are the bilinear interaction strengths, where $K_{ij} = -K_{ji}$, and $J_{1234}$ is the quartic interaction strength.

In the matrix form, this can be written as

$$H = A^\dagger (iM)A + J_{1234}\prod_{i=1}^{4}\gamma_i\,; \qquad A = \begin{pmatrix} \gamma_1 \\ \gamma_2 \\ \gamma_3 \\ \gamma_4 \end{pmatrix}, \tag{27}$$

where $M$ is a $4\times 4$ skew symmetric matrix with six unique non-zero entries, i.e, $K_{ij}$'s. Eigenvalues of the skew-symmetric matrices come in pairs: $\pm\lambda_1, \pm\lambda_2$ with complex raising and lowering operators $f$ and $f^\dagger$ as their eigenstates.

The eigenvectors of $M$ define the basis transformation $U$ from the MZM to the complex fermion representation, i.e, from $\gamma$'s to $f$, $f^\dagger$. The Hamiltonian corresponding to the bilinear MZM interactions, $H_{bl}$ transforms as

$$H_{bl} = B^\dagger \tilde{M} B\,; \qquad B = UA = \begin{pmatrix} f_1 \\ f_1^\dagger \\ f_2 \\ f_2^\dagger \end{pmatrix}, \tag{28}$$

where $U$ is an unitary basis transformation matrix from the MZM basis to the complex fermionic basis and $\tilde{M}$ is the transformed bilinear interaction matrix

$$\tilde{M} = U(iM)U^{-1} = \begin{pmatrix} \lambda_1 & 0 & 0 & 0 \\ 0 & -\lambda_1 & 0 & 0 \\ 0 & 0 & \lambda_2 & 0 \\ 0 & 0 & 0 & -\lambda_2 \end{pmatrix}. \tag{29}$$

The two quasiparticle energies are

$$\lambda_{1,2} = \frac{1}{2}\sqrt{\frac{s \pm \sqrt{s^2 - 4d^2}}{2}}, \tag{30}$$

where $s = K_{12}^2 + K_{13}^2 + K_{14}^2 + K_{23}^2 + K_{24}^2 + K_{34}^2$ and $d = K_{14}K_{23} - K_{13}K_{24} + K_{12}K_{34}$. Thus, we can write the bilinear interaction Hamiltonian in this basis as

$$\begin{aligned} H_{bl} &= B^\dagger \tilde{M} B \\ &= \lambda_1(f_1^\dagger f_1 - f_1 f_1^\dagger) + \lambda_2(f_2^\dagger f_2 - f_2 f_2^\dagger) \\ &= \lambda_1(2n_1 - 1) + \lambda_2(2n_2 - 1), \end{aligned} \tag{31}$$

where $n_i$'s are the quasiparticle number operators. $n_i$ is 0 or 1 corresponding to the filled or empty quasiparticle states. Similarly, we can also write the quartic interaction in this basis as

$$H_{1234} = u\gamma_1\gamma_2\gamma_3\gamma_4 = -u(f_1^\dagger f_1 - f_1 f_1^\dagger)(f_2^\dagger f_2 - f_2 f_2^\dagger). \tag{32}$$

The total Hamiltonian in the complex fermion representation is therefore

$$\begin{aligned} H &= \lambda_1(2n_1 - 1) + \lambda_2(2n_2 - 1) \\ &\quad - u(2n_1 - 1)(2n_2 - 1), \end{aligned} \tag{33}$$

where $\{f_i, f_j^\dagger\} = 2\delta_{ij}$ and $f_i^\dagger f_i = 2n_i$.

Thus, we can write the energy levels of this Hamiltonian corresponding to states $|00\rangle$, $|10\rangle$, $|01\rangle$, $|11\rangle$ in the form $|n_1 n_2\rangle$ as

$$E_1^e = \epsilon_0 - \lambda_1 - \lambda_2 - u, \tag{34}$$

$$E_1^o = \epsilon_0 + \lambda_1 - \lambda_2 + u, \tag{35}$$

$$E_2^e = \epsilon_0 + \lambda_1 + \lambda_2 - u, \tag{36}$$

$$E_2^o = \epsilon_0 - \lambda_1 + \lambda_2 + u, \tag{37}$$

where $E^e$'s are the eigenvalues of even parity states, i.e, $|00\rangle$ and $|11\rangle$ and $E^o$'s are eigenvalues of the odd parity states, i.e., $|10\rangle$ and $|01\rangle$. $\epsilon_0$ is a constant shift in the energies.

If we know the spectrum of the eigenstates, we can obtain the Hamiltonian parameters in the complex fermion representation, $\lambda_{1,2}$ and $u$, using the linear transformation

$$\lambda_1 = (-E_1^e + E_2^e + E_1^o - E_2^o)/4, \tag{38}$$

$$\lambda_2 = (-E_1^e + E_2^e - E_1^o + E_2^o)/4, \tag{39}$$

$$u = (E_1^o + E_2^o - E_1^e - E_2^e)/4. \tag{40}$$

We note that multiple Hamiltonians in the MZM representation result in the same eigenspectrum and hence connect to the same complex representation because multiple sets of bilinear interactions $K_{ij}$ map onto the same $\lambda_{1,2}$. However, there are two important features of the mapping betwen the representations: (1) the quartic interaction $u$ is identical in both representations (same in magnitude except for a sign change); (2) we are interested in having all bilinear interactions in the MZM representation being zero; Since $\lambda_1 = \lambda_2 = 0$ if and only if $K_{ij} = 0 \,\forall\, i,j$ we can verify that we have nulled the bilinear interactions by verifying that $\lambda_1 = \lambda_2 = 0$.

## C  Majorana mode wave function overlap optimization

Following discussion in Section 5.1, we look for optimal points in the parameter space of $\mu_{\text{t1}}$, $\mu_{\text{t2}}$, $\mu_{\text{nt}}$, $t$ and $\Delta$, at which the overlap integral of Majorana modes cancel out, i.e., points at which $E_{|11\rangle} - E_{|00\rangle} \simeq 0$. We perform a global search within the total parameter space to look for the global minima of $\left|E_{|11\rangle} - E_{|00\rangle}\right|$. To visualize this optimization process better, we have plotted $E_{|11\rangle} - E_{|00\rangle}$ as a function of two parameters, keeping others fixed as shown in Fig. 12. This shows how each parameter contributes in the optimization process.

## D  Spatial distribution of the MZM's in the Kitaev model

Following the discussion in Section 5.1, overlap integral of MZM wave functions in a 1D Kitaev chain (with no interactions) can be tuned to zero at a point where lowest four many-body states are almost degenerate. Here we show that indeed such MZM wave function overlap can be seen at an optimal point (which we find by following method in Section 5.1.1) as shown in Fig. 13. To see how these Majorana modes extend throughout the 1D chain, we find weights of the left and right polarized Majorana modes on $\gamma$'s per site.

Defining Majorana creation and annihilation operators corresponding to the left and right Majorana modes - $\tilde{\gamma_x}$ and $\tilde{\gamma_y}$ as:

$$\tilde{\gamma_x}|\psi_0\rangle = |\psi_1\rangle, \qquad \tilde{\gamma_y}|\psi_0\rangle = i|\psi_1\rangle, \tag{41}$$

where $\psi_0$ and $\psi_1$ are the ground states. We can find the weights of $\tilde{\gamma_x}$ and $\tilde{\gamma_y}$ on each site as:

$$\tilde{\gamma_x} = \sum_j \alpha_j^x \gamma_j, \qquad \tilde{\gamma_y} = i\sum_j \alpha_j^y \gamma_j, \tag{42}$$

where $\alpha_j^x$ and $\alpha_j^y$ are the weights of $\tilde{\gamma_x}$ and $\tilde{\gamma_y}$ operators on the $\gamma$'s per site:

$$\alpha_j^x = \langle\psi_1|\gamma_j|\psi_0\rangle, \qquad \alpha_j^y = \langle\psi_1|\gamma_j|\psi_0\rangle. \tag{43}$$

At the optimal point for $U = 0$, $\tilde{\gamma_x}$ and $\tilde{\gamma_y}$ for $\psi_0 \equiv \psi_{00}$ and $\psi_1 \equiv \psi_{01}$ is shown in Fig. 13(a), and for $\psi_0 \equiv \psi_{00}$, $\psi_1 \equiv \psi_{10}$ is shown in Fig. 13(b).

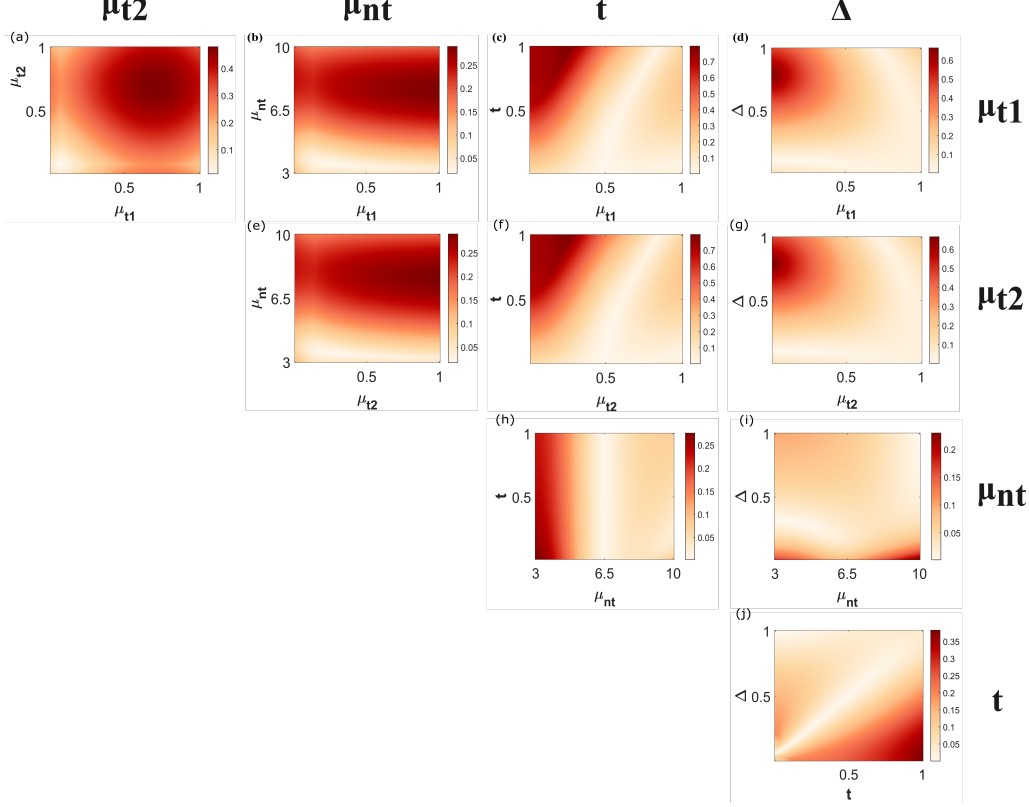

Figure 12: $E_{|11\rangle} - E_{|00\rangle}$ is a function of parameters $\mu_{t1}$, $\mu_{t2}$, $\mu_{nt}$, $t$ and $\Delta$. Colormap showing dependence of $E_{|11\rangle} - E_{|00\rangle}$ on a pair of parameters in each sub-figure (keeping others fixed). The fixed parameters in the sub-figures for $N = 10$ site Kitaev chain with non-topological segment: $n_{nt_1} = 4$; $n_{nt_2} = 7$ are as follows. (a), $t = 1$, $\Delta = 0.5$, $\mu_{nt} = 5$. (b), $\mu_{t2} = 0.1$, $t = 1$, $\Delta = 0.5$. (c), $\mu_{t2} = 0.1$, $\mu_{nt} = 5$, $\Delta = 0.5$. (d), $\mu_{t2} = 0.1$, $\mu_{nt} = 5$, $\Delta = 0.5$. (e), $\mu_{t1} = 0.1$, $t = 1$, $\Delta = 0.5$. (f), $\mu_{t1} = 0.1$, $\mu_{nt} = 5$, $\Delta = 0.5$. (g), $\mu_{t1} = 0.1$, $\mu_{nt} = 5$, $t = 1$. (h), $\mu_{t1} = 0.1$, $\mu_{t2} = 0.1$, $\Delta = 0.5$. (i), $\mu_{t1} = 0.1$, $\mu_{t2} = 0.1$, $t = 1$. (j), $\mu_{t1} = 0.1$, $\mu_{t2} = 0.1$, $\mu_{nt} = 5$.

# E  Finding optimal point using a global search algorithm

Following Section 5.1.1, we search for global minima of the function $E_{|11\rangle} - E_{|00\rangle}$ depending on the parameters $\mu_{t1}$, $\mu_{t2}$, $\mu_{nt}$, $t$ and $\Delta$, i.e, $\left| E_{|11\rangle} - E_{|00\rangle} \right| \simeq 0$. We use a MATLAB global search algorithm [49] which performs multiple parallel searches (using a nonlinear programming solver - 'fmincon' [50]) through the parameter space with different start points to find multiple local minima and then finalizes at a global minimum. The search ranges that were used are: $\{\mu_{t1}, \mu_{t2}, t, \Delta\} \in [0.1, 1]$ and $\mu_{nt} \in [3, 10]$. Optimization stopping criteria: Function tolerance $\sim e^{-10}$, Max Iterations: 3000.

## E.1  Three complex fermion, i.e, six MZM case

We search for global minima of the function $\left| E_{|111\rangle} - E_{|000\rangle} \right|$ dependent on the parameters $\mu_{t1}$, $\mu_{t2}$, $\mu_{t3}$, $\mu_{nt1}$, $\mu_{nt2}$, $t$ and $\Delta$. The search algorithm is same as the two complex fermion case, i.e., MATLAB global search algorithm [49]. The search ranges that were used are: $\{\mu_{t1}, \mu_{t2}, \mu_{t3}, t, \Delta\} \in [0.1, 1]$, $\{\mu_{nt1}, \mu_{nt2}\} \in [3, 10]$.

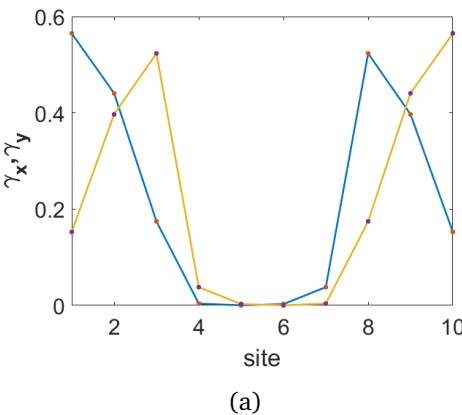 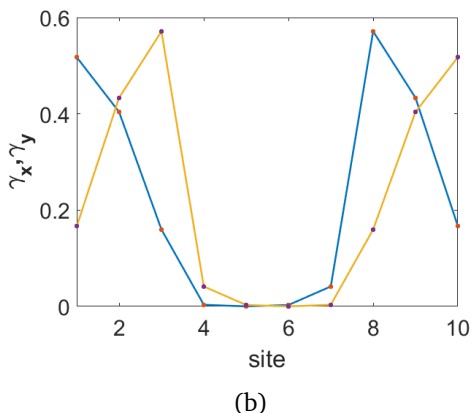

(a)                (b)

Figure 13: Left and right MZM's ($\tilde{\gamma_x}$ (blue) and $\tilde{\gamma_y}$ (yellow)) for states $\psi_1 = |01\rangle$ (in (a)) and $\psi_1 = |10\rangle$ (in (b)). $\psi_0 = |00\rangle$. The parameters are the optimal point at which $\left|E_{|11\rangle} - E_{|00\rangle}\right| \simeq 0$: $t = 0.4025$, $\Delta = 0.2167$, $\mu_{t1} = 0.4832$, $\mu_{t2} = 0.4832$, $\mu_{nt} = 8.5364$ for $N = 10$ site Kitaev chain with non-topological segment: $n_{nt_1} = 4, n_{nt_2} = 7$.

## F   Genetic Algorithm search

It starts by generating a random population of individuals (vectors of double type). The next generation of population is selected based on elites (individuals with best fitness to selectivity criteria), crossover (combining certain parents to create children) and mutation (random modification to some parents in the population). We use MATLAB's Genetic Algorithm Toolbox [51] to solve our problem. These are the criterion that we use in our code: Creation function for creating initial population is the MATLAB default option. Selection function that decides how to select next generation of population - 'selectiontournament'. Crossover function - 'crossoverscattered'. Mutation Function - 'mutationadaptfeasible'. Hybrid function that refines search once Genetic Algorithm search terminates - 'fmincon' [50]. Optimization stopping criteria: 'FunctionTolerance' $\sim e^{-6}$, 'MaxStallGenerations' (controls the number of steps the Genetic Algorithm looks over to see whether it is making progress) - 100.

Parameters to be optimized are $\mu_{t1}$, $\mu_{t2}$, $\mu_{nt}$, $t$, $\Delta$ and $U$. Range of search: $\{\mu_{t1}, \mu_{t2}, t, \Delta\} \in [0.1, 1]$, $\mu_{nt} \in [3, 10]$, $U \in [0.001, 1]$.

### F.1   Three complex fermion, i.e, six MZM case

Genetic Algorithm criterion used in the code are same as the two-complex-fermion case. Parameters to be optimized are $\mu_{t1}$, $\mu_{t2}$, $\mu_{t3}$, $\mu_{nt1}$, $\mu_{nt2}$, $t$, $\Delta$ and $U$. Range of search: $\{\mu_{t1}, \mu_{t2}, \mu_{t3}, t, \Delta\} \in [0.1, 1]$, $\{\mu_{nt1}, \mu_{nt2}\} \in [3, 15]$, $U \in [0.01, 1]$.

## G   Exploring other types of interactions

Besides the non-linear interaction term $H_{\text{nl-int}} = U \sum_{i<j} c_i^\dagger c_i c_j^\dagger c_j$, we have explored other interaction terms like $H_{\text{int}} = U \sum_i c_i^\dagger c_i c_{i+1}^\dagger c_{i+1}$, $U \sum_i c_i^\dagger c_i c_{i+2}^\dagger c_{i+2}$, $U \sum_{i<j} c_i^\dagger c_i c_{i+3}^\dagger c_{i+3}$ and $U \sum_{i<j} c_i^\dagger c_i c_{i+4}^\dagger c_{i+4}$ (with interaction strength $U$) shown in Fig. 16. We sweep across the value of $U$, fixing other parameters at the optimal point (described in Section 5.1). As we follow the trajectory of $\lambda$'s and $u$ in the sub-figures, we do not find points where $\lambda_{1,2} \simeq 0$ and $u$ is non-zero, thus showing that no SYK point exists within the plausible range of $U$.

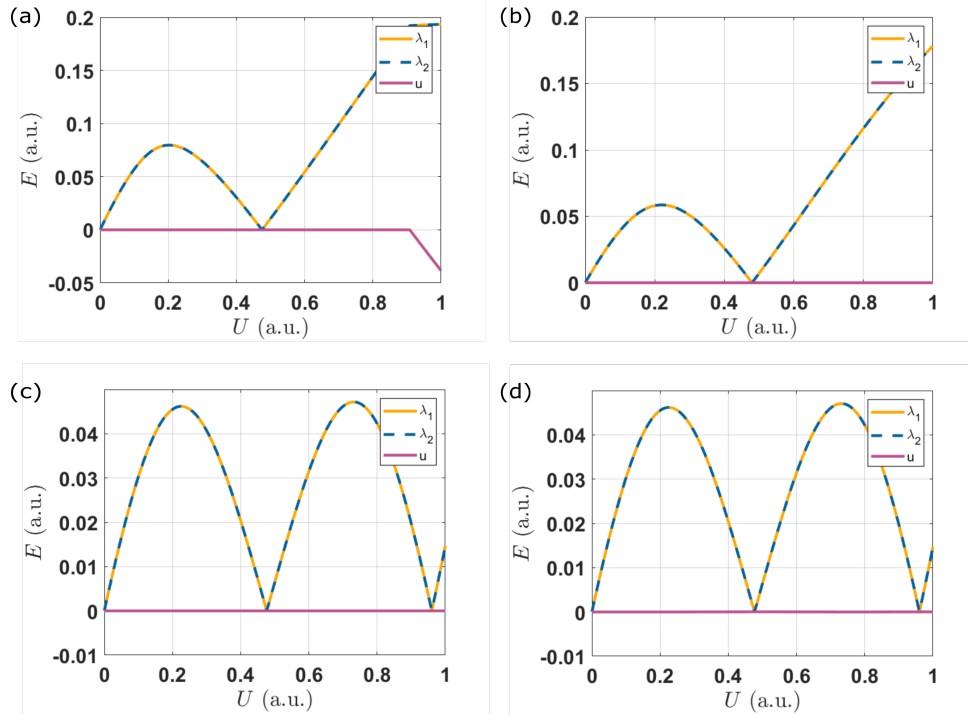

Figure 14: Other interaction terms we checked are - (a) $H_{\text{int}} = U \sum_i c_i^\dagger c_i c_{i+1}^\dagger c_{i+1}$, (b) $H_{\text{int}} = U \sum_i c_i^\dagger c_i c_{i+2}^\dagger c_{i+2}$, (c) $H_{\text{int}} = U \sum_{i<j} c_i^\dagger c_i c_{i+3}^\dagger c_{i+3}$, (d) $H_{\text{int}} = U \sum_{i<j} c_i^\dagger c_i c_{i+4}^\dagger c_{i+4}$. All other parameters are tuned to the optimal point at which $\left| E_{|11\rangle} - E_{|00\rangle} \right| \simeq 0$: $t = 0.4025$, $\Delta = 0.2167$, $\mu_{\text{t1}} = 0.4832$, $\mu_{\text{t2}} = 0.4832$, $\mu_{\text{nt}} = 8.5364$ for $N = 10$ site Kitaev chain with non-topological segment: $n_{\text{nt}_1} = 4, n_{\text{nt}_2} = 7$.

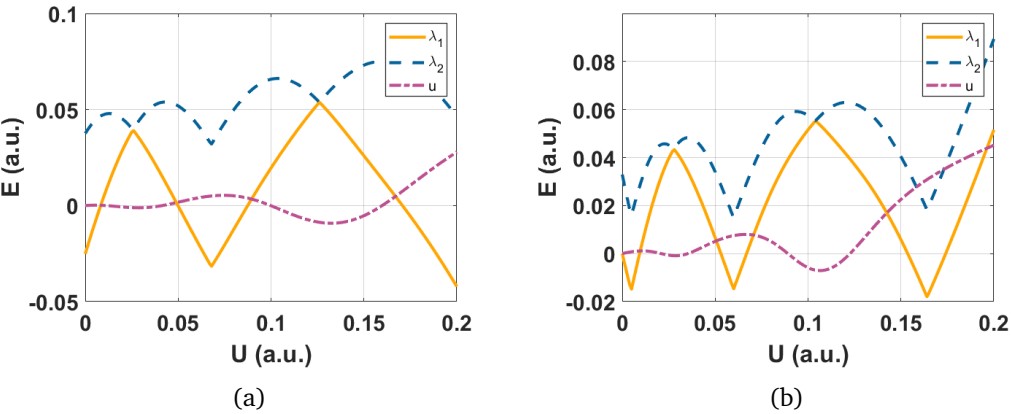

Figure 15: Changing parameters from the optimal point shows that $\lambda's$ no longer overlap and follow separate trajctories (solid yellow and dashed blue line). Parameters used in this figure are as follows. (a), $N = 10$ site Kitaev chain with non-topological segment: $n_{\text{nt}_1} = 4; n_{\text{nt}_2} = 7$; $t = 0.4049$, $\Delta = 0.2207$, $\mu_{\text{t1}} = 0.5513$, $\mu_{\text{t2}} = 0.347$, $\mu_{\text{nt}} = 7.7363$. (b), $N = 10$ fermion Kitaev chain with non-topological segment: $n_{\text{nt}_1} = 4; n_{\text{nt}_2} = 7$; $t = 0.2801$, $\Delta = 0.1278$, $\mu_{\text{t1}} = 0.3544$, $\mu_{\text{t2}} = 0.4324$, $\mu_{\text{nt}} = 8.515$.

# H  Lambda's follow different path beyond the optimal points

As shown in Figs. 5, 6, 7, $\lambda_1 = \lambda_2$ as we tune the non-linear interaction strength $U$. This is a consequence of starting at either an optimal point or an SYK point. In Fig. 15, we show that if we change the parameters slightly, then the $\lambda$'s follow separate paths.

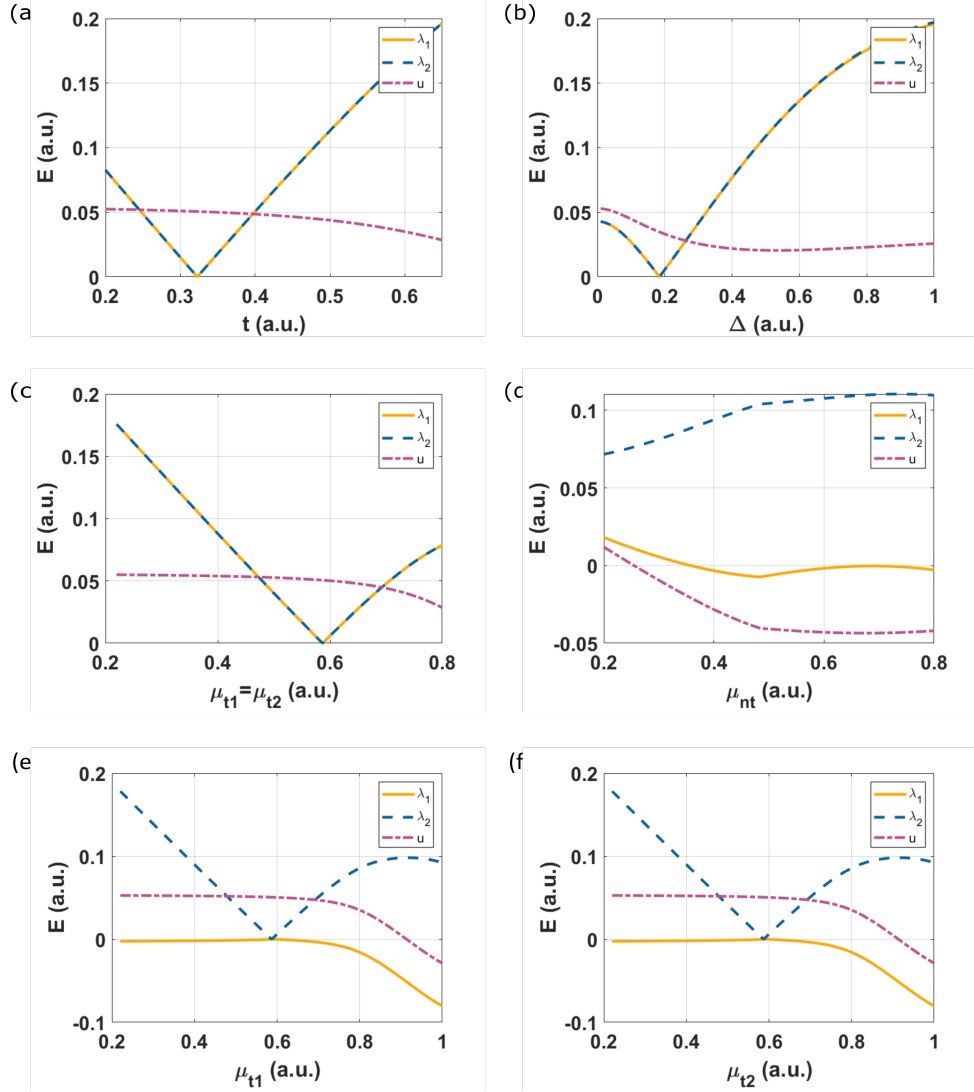

Figure 16:  We plot interaction strengths $\lambda_1,\lambda_2,u$ as a function of parameters of Hamiltonian in Eq. 11. The interaction strengths are plotted as a function of one parameter keeping the others fixed to the SYK point paramteric values mentioned in Fig. 6: We plot $\lambda_1,\lambda_2,u$ as a function of Fig.(a). $t$, Fig.(b). $\Delta$, Fig.(c). $\mu_{t1}$ and $\mu_{t2}$ where $\mu_{t1} = \mu_{t2}$ are varied together, Fig.(d). $\mu_{nt}$, Fig.(e). $\mu_{t1}$,Fig.(d). $\mu_{t2}$.

# I  Interaction strengths as a function of all parameters

We plot interaction strengths $\lambda_1,\lambda_2,u$ as a function of parameters in Hamiltonian in Eq. 11 as a function of one parameter ($\mu_{t1},\mu_{t2},t,\Delta$) and keeping the other parameters fixed at the SYK point mentioned in Fig. 6 (except for Fig. (c) where $\mu_{t1} = \mu_{t2}$ are varied together), shown

in Fig. 16. As seen from Fig. 16 (a), (b) and (c), we find SYK points as function of $t$, $\Delta$ and $\mu_{t1} = \mu_{t2}$, however we couldn't tune to an SYK point as a function of $\mu_{t1}$, $\mu_{t2}$ and $\mu_{nt}$ when swept through independently, shown in Fig. 16 (d), (e) and (f). Hence we show that it's possible to find SYK points if we sweep through the gate-tunable Kitaev chain parameters and keep $U$ fixed, since $U$ can be difficult to tune in experiments.

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
