# Peer review of "A proposal to extract and enhance four-Majorana interactions in hybrid nanowires"

_SciPost Physics, doi:SciPost Phys. 13, 120 (2022)_

## Round 1 · Referee Report · Anonymous · 2021-11-26

Strengths
1- Clearly written
2-Physically sound
Weaknesses
1-Limited applicability
Report
In this paper, the authors study a system of four Majorana modes, which can be written as two complex fermions. Their goal is to find a way to build the SYK model using these four-Majorana-systems as building blocks. By creating a model of an interacting Kitaev chain and including interactions, they show that by tuning several parameters they are able to minimize the bilinear terms in the low-energy Hamiltonian, a requirement for the SYK model. They demonstrate their procedure with numerics for a 4-Majorana and 6-Majorana model.
This paper meets all of the general acceptance criteria: it is clearly written and appropriately references and cites its sources. However, it must also meet one of SciPost's expectations, which I feel at the moment it currently does not. The paper hopes its calculations may be applied to simulate the SYK model, which would be an appropriate criterion for publication, but there are several important considerations that need to be taken into account for this to happen:
First, the paper only considers 4 and 6 Majoranas, or 2 and 3 complex fermions. As the number of fermion grows, the Hilbert space will grow exponentially (and number of bilinears quadratically), while the number of parameters available to tune will grow linearly. I suspect that this minimization procedure will not work as N grows large, which is the regime of SYK; there are far too many bilinears to minimize.
Second, the SYK model not only requires the absence of bilinear terms, but also a random four-body interaction between all Majorana modes. It is insufficient to prove that the four-body term is nonzero. As interactions typically decay with distance, it is difficult to see how such a term will arise in the Kitaev chain suggested in the paper.
Third, the parameters tuned in the model include the pairing strength and the interaction. How are these tuned in actual physical systems? In addition, the authors mention that tunneling spectroscopy will be able to determine the energy parameters in Eq. 5, but how does one determine which energy corresponds to which state E_{a,b,c,...}, especially in the presence of superconductivity, where total fermion number is not a good quantum number? The authors show this is possible for 2 complex fermions, with only knowledge of the parity of the states to determine the parameters \lambda and u. But is this true for higher numbers of fermions?
In summary, the paper is well-written and clear. However, in order to have the necessary relevance, it needs to show the 4-Majorana analysis can be extended to the SYK model (with large numbers of Majoranas). If the appropriate points above are addressed, I would be happy to recommend publication.
Requested changes
I address specific points by section.
III.
1-From an experimental spectrum of the energy eigenstates, how does one determine which occupations a,b,c,... in E_{a,b,c...} a certain energy level corresponds to, for more than 2 complex fermions?
IV.
2-How physical is the interaction term in Eq. 12? It does not decay with distance, as most interactions should. By not decaying with distance, it potentially can generate the all-to-all interactions required for SYK, but such interactions should not occur in nature.
V.
3-How does one tune \mu, \Delta, and U in experiment? In general, are there enough parameters to tune in an experimental model to reach the SYK point?
General
4-Is the analysis for the 4-Majorana model extensible to N-Majorana models, where N is very large? Can all bilinears be suppressed by tuning appropriate terms, and can the four-body interaction generated be all-to-all and random?
Author: Sergey Frolov on 2022-05-24 [id 2522]
(in reply to Report 1 on 2021-11-26)
Report1
In this paper, the authors study a system of four Majorana modes, which can be written as two complex fermions. Their goal is to find a way to build the SYK model using these four-Majorana-systems as building blocks.
The goal of this work in NOT to build the SYK model. It is to envision experiments capable of detecting four-Majorana interactions which are the building blocks of the SYK model. This work is inspired by the ideas of SYK but it addresses a separate and well-defined problem. In a system relevant to experiments, represented by the Kitaev chain here, how can the four-Majorana interactions be separated from two-Majorana interactions and how can these types of interactions be separately tuned? These questions are of independent value - not every paper must be another SYK proposal. In fact, previous works have not focused on the questions we pose here.
By creating a model of an interacting Kitaev chain and including interactions, they show that by tuning several parameters they are able to minimize the bilinear terms in the low-energy Hamiltonian, a requirement for the SYK model. They demonstrate their procedure with numerics for a 4-Majorana and 6-Majorana model. This paper meets all of the general acceptance criteria: it is clearly written and appropriately references and cites its sources. However, it must also meet one of SciPost's expectations, which I feel at the moment it currently does not. The paper hopes its calculations may be applied to simulate the SYK model, which would be an appropriate criterion for publication, but there are several important considerations that need to be taken into account for this to happen:
We disagree. The paper is not about building the SYK model - see above. It is a proposal for how experiments could be conducted in systems not too far from the experimental state of the art. Please see the comment above. We have also added a new section VIII where we lay out how the findings of this work can be used in a future experiment, in a step-by-step fashion. We hope this section makes clear the experimental relevance of this work.
First, the paper only considers 4 and 6 Majoranas, or 2 and 3 complex fermions. As the number of fermion grows, the Hilbert space will grow exponentially (and number of bilinears quadratically), while the number of parameters available to tune will grow linearly. I suspect that this minimization procedure will not work as N grows large, which is the regime of SYK; there are far too many bilinears to minimize.
This work is an experimental proposal for estimating and enhancing four-Majorana interactions in a toy system of 4 or 6 Majoranas. If such an experiments is attempted, this paper will offer a template on how to evaluate the presence of these interactions and on how to enhance them. This paper DOES NOT deal with N Majorana or the SYK models. There are already plenty of papers dealing with that.
Second, the SYK model not only requires the absence of bilinear terms, but also a random four-body interaction between all Majorana modes. It is insufficient to prove that the four-body term is nonzero. As interactions typically decay with distance, it is difficult to see how such a term will arise in the Kitaev chain suggested in the paper.
In this paper we consider a system of 4 or 6 MZM in order to demonstrate how the strength of four-Majorana interactions can be estimated in real experiments. The notion of random coupling does not apply to our paper. It is relevant to the SYK model which is the inspiration for our work. But this work focuses on different questions and does not focus on the realization of the SYK model.
Third, the parameters tuned in the model include the pairing strength and the interaction. How are these tuned in actual physical systems?
These are hard to tune in experiment. We plot what would happen for a range of these parameters because the value is not known a priori for a real systems. Tuning of these parameters can be achieved by changing the design of the device, e.g. location of MZM and the amount of screening by electrode placement, by controlling density/doping, suspending wires and other means.
In addition, the authors mention that tunneling spectroscopy will be able to determine the energy parameters in Eq. 5, but how does one determine which energy corresponds to which state E_{a,b,c,...}, especially in the presence of superconductivity, where total fermion number is not a good quantum number? The authors show this is possible for 2 complex fermions, with only knowledge of the parity of the states to determine the parameters \lambda and u. But is this true for higher numbers of fermions?
For 2 or 3 complex fermions, the idea is that there will be the right number of subgap resonances in tunneling measurements and they will behave as expected for different parity states, e.g. they will move up/down in energy in magnetic field as expected - this is one way of identifying the parity in Andreev bound state measurements. Parity can also be read-out real-time in more advanced experiments (‘Majorana qubits’). All of this assumes high quality samples and a large degree of control, but at the level of proposed future experiments this is valid. As discussed, we do not consider higher number of fermions, given that experiments struggle with 1 fermion at this time. Going from 4 to 6 MZM was meant to demonstrate that this is possible, in principle, though the complexity grows.
In summary, the paper is well-written and clear. However, in order to have the necessary relevance, it needs to show the 4-Majorana analysis can be extended to the SYK model (with large numbers of Majoranas). If the appropriate points above are addressed, I would be happy to recommend publication.
We appreciate the comments of the referee except we disagree on the relevance of higher N cases. The case of N=4 MZM is already way beyond the experimental state of the art. Our paper is a theory/numerical contribution that balances the experimental realities with inspiration drawn from the SYK model. This is where the significance of the work is found. It is not just another magic SYK proposal where everything just cancels out. The formalism and numerics developed here will guide future experiments when they try to identify the presence of 4-Majorana interactions in real samples.
Requested Changes I address specific points by section. III. 1-From an experimental spectrum of the energy eigenstates, how does one determine which occupations a,b,c,... in E_{a,b,c...} a certain energy level corresponds to, for more than 2 complex fermions?
We point out that the paper focues primarily on N=2 complex fermions which is already an awesome experimental challenge. For N>2 complex fermion, determining individual energy levels by their occupation number may not be possible. However, tunneling spectroscopy data can be used to determine energy level spacings between even and odd parity states which will give us the interaction strengths. Although, it could be tricky to analyze the data when degeneracies, i.e., at energy level crossings appear. IV.
2-How physical is the interaction term in Eq. 12? It does not decay with distance, as most interactions should. By not decaying with distance, it potentially can generate the all-to-all interactions required for SYK, but such interactions should not occur in nature.
This interaction term in Eq. 12 is a coulomb interaction that exists in nature although maybe with minimal strength in the case of a 1D nanowire. We can effectively simulate its non-local nature by modifying the geometry of our model, e.g., in the case where we have two different nanowires, hosting a pair of MZMs, placed parallel to each other instead of having one nanowire with three topo, non-topo, topo segments. We have added a discussion on this in Section IV.B of the paper.
V.
3-How does one tune \mu, \Delta, and U in experiment? In general, are there enough parameters to tune in an experimental model to reach the SYK point?
\mu can be tuned with gates. \Delta may be harder to tune in situ but could be possible in principle. \Delta can also be pre-selected by the choice of material that we use for superconductor-semiconductor nanowire. U could be harder to tune in a 1D wire but can be controlled by modifying the geometry of the device, we have added this discussion in Section IV.B. Yes, there are enough parameters to tune and we do not need to tune all the parameters simultaneously to reach the SYK point, we have added a discussion in the paper explaining this in Appendix I. We have also added Section VIII in the paper where we discuss the experimental protocol to reach SYK point for the 4-MZM case in detail.
General
4-Is the analysis for the 4-Majorana model extensible to N-Majorana models, where N is very large? Can all bilinears be suppressed by tuning appropriate terms, and can the four-body interaction generated be all-to-all and random?
In this paper we focus on a small N model and mostly on the 4-MZM model. We think that a 1D Kitaev chain model will not be suitable for a large N-Majorana model, however this might be possible with modification in geometry. In principle, in such a model all bilinears can be suppressed by following the same eigenspectral analysis as discussed in the paper.
We can also introduce randomness by adding a random strength to the non-local interaction term. However, in this paper, we focus on building a 4-MZM interaction system and not an SYK system, hence we do not add the condition for randomness in our model.
Author: Sergey Frolov on 2022-05-24 [id 2524]
(in reply to Report 3 on 2021-12-21)Report3
This is not correct. We do not propose the realization of the SYK model. We propose how 4-MZM interactions can be extracted experimentally in realistic systems, and how they can be enhanced relative to 2-MZM interactions. This is a question of relevance to ongoing experiments, it is of independent value. It is also crucial for any future attempts at realizing the full SYK model - because if you don’t have detectable 4-MZM terms then you cannot build the SYK model…
This is true for any SYK proposal. At least we offer a way to quantify 4-MZM terms however small they may be….

---

## Round 1 · Referee Report · Anonymous · 2021-12-8

Report
The Authors characterize the four-Majorana interaction in a system hosting a few Majorana zero modes by its energy spectrum.
The central result of the paper is that for a one-dimensional Kitaev chain hosting four Majoranas in the presence of all-to-all density-density interaction, one can suppress the bilinear terms and maximize the four-Majorana interaction in the low-energy theory. To do so, the Authors systematically tune the original Hamiltonian using the relation between the coupling constants in the low-energy theory to the four lowest energy levels.
The goal is to provide a minimal building block for the experimental realization of the Sachdev-Ye-Kitaev model - a system with a four-Majorana interaction with suppressed bilinear terms. The Authors show a way to eliminate the bilinear contribution in a few-Majorana system.
The paper is very well structured and written clearly.
However, I think some issues would need to be addressed before suggesting the paper for publication:
1) What is the origin of the non-local interaction term (12)? Does it appear as a low-energy projection of density-density interaction with the Coulomb potential?
Are there any conditions for the nanowire parameters to provide this type of long-range interaction?
2) The SYK model describes the collective behavior of large-$N_\gamma$ randomly interacting zero modes. Providing that the interaction (12) can be made non-local for $N_\gamma>4$, is there a way to make the resulting four-Majorana interaction random in the suggested setup?
3) According to Fig. 8a, the amplitudes of the two-body interaction for $N_\gamma = 6$ are equal $u_1=u_2=u_3$. Does it imply equal $J_{ijkl}$ in the Hamiltonian (13)? If yes, can it be violated by adding disorder?
Apart from that, I would like to ask for two brief clarifications:
- The energy characteristic curves Figs. 4-6, 8,9 are Energy vs $U$, where $U$ is the strength of the original two-body interaction (12). However, in Section VII the Authors mention that $U$ is the hardest to tune among the parameters of the problem and, hence, the hardest to variate in the experiment. I am wondering if similar energy characteristic curves can be potentially done for a different parameter along the x-axis?
- In the second paragraph of Section V.B. the Authors write: "Our optimization problem now consists of a six parameter space $\lbrace\mu_{t1}, \mu_{t2}, t, \Delta, U\rbrace$ ..." Though, there are only five parameters listed. Is there an extra parameter?
In summary, the paper demonstrates that it is possible to engineer a few-MZM system with four-Majorana interaction and suppressed bilinear terms in one dimension. It definitely meets SciPost's general acceptance criteria.
However, it is not clear to me yet if the results can be extended to a low $N_\gamma$ realization of the SYK model.
As I mentioned above in (1-3), It seems to me that the paper in the current version requires some additions. Namely, adding the discussion on the applicability of the long-range interaction Hamiltonian (12) and a possible mechanism to make the resulting four-Majorana interaction random for $N_\gamma>4$. If these issues can be resolved, then the paper would check off "3. Open a new pathway in an existing or a new research direction, with clear potential for multipronged follow-up work" among Scipost expectations as a novel proposal for realizing the SYK model in one dimension.
Requested changes
1. Adding the discussion on the applicability of the all-to-all density-density Hamiltonian (12).
2. Adding a condition that would allow for random four-Majorana terms at the SYK point.
Author: Sergey Frolov on 2022-05-24 [id 2523]
(in reply to Report 2 on 2021-12-08)
Report2
The Authors characterize the four-Majorana interaction in a system hosting a few Majorana zero modes by its energy spectrum. The central result of the paper is that for a one-dimensional Kitaev chain hosting four Majoranas in the presence of all-to-all density-density interaction, one can suppress the bilinear terms and maximize the four-Majorana interaction in the low-energy theory. To do so, the Authors systematically tune the original Hamiltonian using the relation between the coupling constants in the low-energy theory to the four lowest energy levels. The goal is to provide a minimal building block for the experimental realization of the Sachdev-Ye-Kitaev model - a system with a four-Majorana interaction with suppressed bilinear terms. The Authors show a way to eliminate the bilinear contribution in a few-Majorana system. The paper is very well structured and written clearly. I think some issues would need to be addressed before suggesting the paper for publication: 1) What is the origin of the non-local interaction term (12)? Does it appear as a low-energy projection of density-density interaction with the Coulomb potential?
The interaction is introduced ad-hoc. It can be a density-density interaction - this would be the least exotic possible origin.
Are there any conditions for the nanowire parameters to provide this type of long-range interaction?
This interaction exists in nature although maybe with minimal strength. This would be a problem for all SYK proposals. Here we do not assume a specific strength but propose a way to extract and maximize the strength. Here are some ideas for how to enhance it in practice. We can enhance it in principle by minimizing screening effects from nearby metals. We can also effectively enhance it by modifying the geometry of our model, e.g., in a case where we have two different nanowires hosting a pair of MZMs placed parallel to each other instead of having one nanowire with three topo, non-topo, topo segments, like the figure in the manuscript shows.
2) The SYK model describes the collective behavior of large-Nγ randomly interacting zero modes. Providing that the interaction (12) can be made non-local for Nγ>4, is there a way to make the resulting four-Majorana interaction random in the suggested setup?
Our work does not address the question of large-N MZM systems. Though this would be required to realize SYK eventually, here we ask what would be a practical way to even detect the presence of any 4-MZM interaction in a realistic system? Without a way to estimate and enhance 4-way interactions, over 2-way interactions, there is surely no path to SYK. It is also an experimental question of separate importance - how to study interacting MZM? That being said, yes, we could add randomness to the non-local interaction strength U which will effectively add randomness to the interactions in our system; and then following the method of minimizing bilinear interactions and maximizing quartic interactions as discussed in the paper, we could in principle arrive at the SYK point. However, this could be difficult to achieve in an experimental setup.
3) According to Fig. 8a, the amplitudes of the two-body interaction for Nγ=6 are equal u1=u2=u3. Does it imply equal Jijkl in the Hamiltonian (13)? If yes, can it be violated by adding disorder?
In Fig 8.a, u1,2,3 are the amplitudes of the four body interactions: u12, u13, u23 as described in Section VI.C. u’s being equal in this case does not automatically imply Jijkl are equal since they aren’t one to one related for Nγ=6 case. However, we have not looked at this question since our work has been carried out in the complex fermion basis.
Apart from that, I would like to ask for two brief clarifications: - The energy characteristic curves Figs. 4-6, 8,9 are Energy vs U, where U is the strength of the original two-body interaction (12). However, in Section VII the Authors mention that U is the hardest to tune among the parameters of the problem and, hence, the hardest to variate in the experiment. I am wondering if similar energy characteristic curves can be potentially done for a different parameter along the x-axis?
Yes, we can get similar energy characteristic curves against other parameters such as t, \Delta and when \mu_{t1} and \mu_{t2} are both simultaneously varied along the x-axis. We have added this discussion in Appendix I in the paper.
- In the second paragraph of Section V.B. the Authors write: "Our optimization problem now consists of a six parameter space {μt1,μt2,t,Δ,U}{μt1,μt2,t,Δ,U} ..." Though, there are only five parameters listed. Is there an extra parameter?
We have fixed the typo.
In summary, the paper demonstrates that it is possible to engineer a few-MZM system with four-Majorana interaction and suppressed bilinear terms in one dimension. It definitely meets SciPost's general acceptance criteria. However, it is not clear to me yet if the results can be extended to a low Nγ realization of the SYK model.
As discussed above, this is not a proposal to realize SYK, but to extract and enhance 4-Majorana interactions. This is inspired by SYK, but it is a distinct question of more immediate experimental relevance than SYK itself. As I mentioned above in (1-3), It seems to me that the paper in the current version requires some additions.Namely, adding the discussion on the applicability of the long-range interaction Hamiltonian (12) and a possible mechanism to make the resulting four-Majorana interaction random for Nγ>4. If these issues can be resolved, then the paper would check off "3. Open a new pathway in an existing or a new research direction, with clear potential for multipronged follow-up work" among Scipost expectations as a novel proposal for realizing the SYK model in one dimension. We have added a new section VIII where we lay out how the findings of this work can be used in a future experiment, in a step-by-step fashion. We hope this section makes clear the experimental relevance of this work and how it opens a new pathway per SciPost criteria…
Requested Changes
- Adding the discussion on the applicability of the all-to-all density-density Hamiltonian (12).
We have added a discussion on this in Section IV.B in the paper.
- Adding a condition that would allow for random four-Majorana terms at the SYK point.
As discussed above, the focus of this work is not the aspect of SYK that requires random interactions of many-MZM in quartets, but on a more immediately relevant question - are there any 4-MZM interactions in realistic systems and how can they be detected/enhanced?
In practical terms, because we focus on building a 4-MZM interaction system and not an SYK system, hence we do not add the condition for randomness in our model. There is nothing to randomize in this model.

---

## Round 1 · Referee Report · Anonymous · 2021-12-21

Report
I find that this manuscript is not appropriate for the publication in SciPost. I also doubt that in its present form it may be resubmitted to any other scientific journal.
The idea of authors was to propose a mesoscopic realization of the SYK model. The latter requires a large number N>>1 of Majorana fermions with all-to-all random interaction. (In fact the requirement N>>1 is needed to show the equivalence of SYK low energy physics to the the so-called Schwarzian quantum mechanics, which in turn implies the correspondence to the JT gravity, black-hole physics, many-body chaos and etc.).
The cases discussed by authors are restricted to N=4 and N=6. And from the very idea (splitting of a 1D spin-orbit quantum wire into trivial and topological segments), I do not see a room to scalability of this proposal for larger N. So, in fact, the manuscript discusses (almost) regular spectrum of just few subgap states in the superconducting mesoscopic wire. Increasing the wire length (to get more Majoranas) would also require the usage of very long-ranged Coulomb interaction which unevitibly will be screened by nearby gates which should control the electron concentration in different segments.

---

## Round 2 · Referee Report · Anonymous (Referee 1) · 2022-6-22

Report

The authors have satisfied my questions, and I recommend publication.

---

## Round 2 · Referee Report · Anonymous (Referee 2) · 2022-10-10

Report

The Authors clarified their goal to engineer, tune, and characterize the four-Majorana zero modes (MZM) interaction in the nanowire devices rather than to build the SYK model in a condensed matter platform. The Introduction section was updated accordingly.
In the conclusive Section VII the Authors state that ''... it is indeed possible to design a low $N_γ$ (particularly $N_γ = 4, 6$) SYK model in a 1D nanowire system. It might be possible to extend this model to a $N_γ > 6$ system in future work.'' I believe it would be beneficial for readers to reinstate the initial goal to characterize and control the four-MZM interaction in this Section and specify that achieving scalability for the large-$N$ regime and random character of the SYK interaction is not a part of the paper.
I think the Authors addressed my questions adequately, and I recommend the paper for publication in SciPost.

---

## Round 2 · Author Response

Dear editors and referees,

We have expanded the paper and clarified several points. Please see replies to referees for details.

On the topic of the goal and impact of this work, we would like to clarify - this is not a proposal to implement an SYK model in nanowires. This is a proposal for a future experiment in which the strength of 4-Majorana terms can be extracted for tunneling data, and possibly enhanced, relative to two-Majorana interaction terms. The paper introduces quantities that relate to interaction strengths.

What the paper does not do is study all conditions for the realization of the large N SYK model. This is not the value of this work. Not every paper must do that. There are many important questions to ask and explore that are not full SYK demonstration pathways. The question of how to study interacting Majorana modes is an important one. The proposal paper like this one opens path for future experiments, hence it satisfies SciPost Physics acceptance criteria.

---

## Round 2 · List of Changes

1) Clarified motivation - not full SYK but study interactions

2) Performed additional simulations of tuning via other Kitaev model parameters, rather than the interaction strength U

3) Created a section on the experimental protocol for tunneling experiments

4) Added a figure with a two-wire design which can help enhanece the 4-Majorana term that is non-local

5) Addressed other referee questions

---

## Editorial Decision

published